# Response to immune checkpoint blockade improved in pre-clinical model of breast cancer after bariatric surgery

Laura M Sipe[1], Mehdi Chaib[2], Emily B Korba[1], Heejoon Jo[1], Mary Camille Lovely[1], Brittany R Counts[3], Ubaid Tanveer[1], Jeremiah R Holt[1], Jared C Clements[1], Neena A John[1], Deidre Daria[4], Tony N Marion[4,5], Margaret S Bohm[5], Radhika Sekhri[6†], Ajeeth K Pingili[1], Bin Teng[1], James A Carson[3,7], D Neil Hayes[1,7], Matthew J Davis[8], Katherine L Cook[9], Joseph F Pierre[5,7,10]*, Liza Makowski[1,2,5,7]*

[1]Department of Medicine, Division of Hematology and Oncology, College of Medicine, The University of Tennessee Health Science Center, Memphis, United States; [2]Department of Pharmaceutical Sciences, College of Pharmacy, The University of Tennessee Health Science Center, Memphis, United States; [3]Integrative Muscle Biology Laboratory, Laboratory, Division of Rehabilitation Sciences, College of Health Professions, University of Tennessee Health Science Center, Memphis, United States; [4]Office of Vice Chancellor for Research, University of Tennessee Health Science Center, Memphis, United States; [5]Department of Microbiology, Immunology, and Biochemistry, College of Medicine, The University of Tennessee Health Science Center, Memphis, United States; [6]Department of Pathology, University of Tennessee Health Science Center, Memphis, United States; [7]UTHSC Center for Cancer Research, College of Medicine, The University of Tennessee Health Science Center, Memphis, United States; [8]Department of Surgery, Division of Bariatric Surgery, College of Medicine, The University of Tennessee Health Science Center, Memphis, United States; [9]Department of Surgery, Comprehensive Cancer Center, Wake Forest University School of Medicine, Winston Salem, United States; [10]Department of Nutritional Sciences, College of Agricultural and Life Sciences, University of Wisconsin-Madison, Madison, United States

*For correspondence:
jpierre1@uthsc.edu (JFP);
liza.makowski@uthsc.edu (LM)

Present address: †Montefiore Medical Center. University Hospital for Albert Einstein College of Medicine, New York, United States

Competing interest: The authors declare that no competing interests exist.

**Abstract** Bariatric surgery is a sustainable weight loss approach, including vertical sleeve gastrectomy (VSG). Obesity exacerbates tumor growth, while diet-induced weight loss impairs progression. It remains unknown how bariatric surgery-induced weight loss impacts cancer progression or alters response to therapy. Using a pre-clinical model of obesity followed by VSG or diet-induced weight loss, breast cancer progression and immune checkpoint blockade therapy were investigated. Weight loss by VSG or weight-matched dietary intervention before tumor engraftment protected against obesity-exacerbated tumor progression. However, VSG was not as effective as diet in reducing tumor burden despite achieving similar weight and adiposity loss. Leptin did not associate with changes in tumor burden; however, circulating IL-6 was elevated in VSG mice. Uniquely, VSG tumors displayed elevated inflammation and immune checkpoint ligand PD-L1+ myeloid and non-immune cells. VSG tumors also had reduced T lymphocytes and markers of cytolysis, suggesting an ineffective anti-tumor microenvironment which prompted investigation of immune checkpoint blockade. While obese mice were resistant to immune checkpoint blockade, anti-PD-L1 potently impaired tumor progression after VSG through improved anti-tumor immunity. Thus, in formerly obese mice, surgical weight loss followed by immunotherapy reduced breast cancer burden. Finally, we compared transcriptomic changes in adipose tissue after bariatric surgery from patients and

mouse models. A conserved *bariatric surgery-associated weight loss signature* (BSAS) was identified which significantly associated with decreased tumor volume. Findings demonstrate conserved impacts of obesity and bariatric surgery-induced weight loss pathways associated with breast cancer progression.

## Editor's evaluation

This study investigates how weight loss by bariatric surgery or weight-matched dietary intervention impairs breast cancer growth as well as immunotherapy. This study can potentially provide some therapeutic intervention strategies on combining vertical sleeve gastrectomy and immunotherapy in treating breast cancer.

## Introduction

Obese breast cancer patients, defined as having a BMI greater than 30, have worsened breast cancer prognoses with elevated breast cancer invasion (*Gillespie et al., 2010*; *Neuhouser et al., 2015*), distant metastases (*Ewertz et al., 2011*; *Osman and Hennessy, 2015*; *Mazzarella et al., 2013*), tumor recurrence (*Sestak et al., 2010*; *Biglia et al., 2013*), impaired delivery of systemic therapies (*Anders et al., 2016*; *Ligibel et al., 2014*), and high mortality (*Calle et al., 2003*; *Azrad and Demark-Wahnefried, 2014*; *Lin et al., 2021*). Weight loss interventions focusing on dietary approaches and exercise have demonstrated improved prognoses after a breast cancer diagnosis (*Seiler et al., 2018*; *Ligibel et al., 2017*; *Ligibel et al., 2019*; *Ligibel and Goodwin, 2012*; *Pierce, 2009*). Pre-clinical models support that weight loss through diet or physical activity prior to tumor onset is beneficial to reduce obesity-associated tumor progression (*Friedenreich et al., 2021*; *Lammert et al., 2018*; *Goding Sauer et al., 2019*; *Das et al., 2021*; *Sundaram et al., 2014*). Thus, intentional weight loss prior to tumor onset is a potential intervention to reduce negative cancer outcomes.

Bariatric surgery, also known as metabolic surgery, is an effective intervention for obese patients that leads to stable and sustained weight loss. Bariatric surgery primarily encompasses gastric banding, Roux-en-Y gastric bypass, and vertical sleeve gastrectomy (VSG) (*Bohm et al., 2022*). VSG is currently the least invasive and most common bariatric procedure (*Alalwan et al., 2021*). Patients who receive a VSG have a reduction of 57% excess weight after 2 years, which remains relatively stable out to 10 years post-surgery (*O'Brien et al., 2019*). Remarkably, patients who undergo surgically induced weight loss have a reduction in all-cause mortality up to 60% (*Syn et al., 2021*; *Doumouras et al., 2020*; *Aminian et al., 2022*). Despite promising benefits of weight loss, weight loss regimens are not yet widely adopted in cancer prevention, survivorship, or therapy. Our *premise* is that obese subjects are exposed to chronic inflammation that leads to increased risk of cancer yet induces compensatory immunosuppressive mechanisms or does not achieve a sufficient inflammatory threshold to protect from cancer initiation in a failure of protective immunity. Importantly, bariatric surgery is protective against subsequent risk of developing any cancer by 10–33% (*Aminian et al., 2022*; *Zhang et al., 2020b*). *Feigelson et al., 2020* described the greatest benefit in pre-menopausal estrogen receptor negative cancer in patients after bariatric surgery. A meta-analysis of 11 studies with over 1 million bariatric surgery patients demonstrated a significant 54% reduction in breast cancer incidence compared to BMI-matched controls, regardless of patient age (*Bruno and Berger, 2020*; *Schauer et al., 2017*; *Lovrics et al., 2021*). While there are no specific recommendations for weight loss nor bariatric surgery in patients as a routine cancer prevention approach, the reduction in breast cancer risk associated with weight loss should be further examined using a controlled model system to better understand mechanisms impacting cancer progression and therapeutic efficacy.

Here, to investigate the impacts of obesity and bariatric surgery-induced weight loss on breast cancer progression and response to therapy, we utilized female C57BL/6J mice, which are obesogenic and immune competent. Once obese, mice were subjected to weight loss interventions including bariatric surgery by VSG or dietary intervention as a weight-matched control. Mice not subjected to VSG received a control sham surgery. Mice remaining obese or formerly obese mice that lost weight by surgery or diet were subsequently implanted orthotopically with syngeneic breast cancer cells to determine impacts on tumor progression, burden, and anti-tumor immunity. We found that mice that received the VSG displayed reduced obesity-accelerated breast cancer compared to obese

**eLife digest** As the number of people classified as obese rises globally, so do obesity-related health risks. Studies show that people diagnosed with obesity have inflammation that contributes to tumor growth and their immune system is worse at detecting cancer cells. But weight loss is not currently used as a strategy for preventing or treating cancer.

Surgical procedures for weight loss, also known as 'bariatric surgeries', are becoming increasingly popular. Recent studies have shown that individuals who lose weight after these treatments have a reduced risk of developing tumors. But how bariatric surgery directly impacts cancer progression has not been well studied: does it slow tumor growth or boost the anti-tumor immune response?

To answer these questions, Sipe et al. compared breast tumor growth in groups of laboratory mice that were obese due to being fed a high fat diet. The first group of mice lost weight after undergoing a bariatric surgery in which part of their stomach was removed. The second lost the same amount of weight but after receiving a restricted diet, and the third underwent a fake surgery and did not lose any weight. The experiments found that surgical weight loss cuts breast cancer tumor growth in half compared with obese mice. But mice who lost the same amount of weight through dietary restrictions had even less tumor growth than surgically treated mice.

The surgically treated mice who lost weight had more inflammation than mice in the two other groups, and had increased amounts of proteins and cells that block the immune response to tumors. Giving the surgically treated mice a drug that enhances the immune system's ability to detect and destroy cancer cells reduced inflammation and helped shrink the mice's tumors. Finally, Sipe et al. identified 54 genes which were turned on or off after bariatric surgery in both mice and humans, 11 of which were linked with tumor size.

These findings provide crucial new information about how bariatric surgery can impact cancer progression. Future studies could potentially use the conserved genes identified by Sipe et al. to develop new ways to stimulate the anti-cancer benefits of weight loss without surgery.

sham-treated controls. However, the most effective blunting of tumor progression was detected in weight-matched sham (WM-Sham) controls. Thus, bariatric surgery was effective at reducing tumor burden but not to the same extent as weight-matched controls despite similar weight and adiposity loss between the two groups. A potential mediator limiting the impacts of weight loss on tumor progression after VSG was elevated IL-6, which upregulates the checkpoint ligand, programmed death ligand 1 (PD-L1) on myeloid and non-immune cells, and reduced CD8+T cell content in tumors uniquely in VSG-treated mice. Thus, we next determined if immune checkpoint blockade (ICB) after VSG could improve tumor outcomes. We report that in mice after VSG, anti-PD-L1 was efficacious to reduce breast cancer progression comparable to burdens detected in lean controls, while obese mice were resistant to anti-PD-L1. Finally, using transcriptomic analysis of adipose tissue after bariatric surgery from both patients and mouse models, we identified a conserved *bariatric surgery-associated weight loss signature* (BSAS) that significantly associated with decreased tumor volume. In sum, our study contributes critical observations regarding the impacts of obesity and bariatric surgery-induced weight loss on breast cancer progression and response to immunotherapy that are relevant to this rapidly emerging area of research and medicine.

## Results

### Surgical and dietary weight loss interventions reduced weight to the same extent

To quantify impacts of bariatric surgery on cancer progression, weight loss was induced prior to tumor implantation (study design, *Figure 1A*). Female C57BL/6J mice were weaned onto low fat diet (LFD) to remain lean or onto high fat diet (HFD) to become obese. After 16 weeks on diet, HFD-fed mice displayed marked diet-induced obesity (DIO, *Figure 1B*). A subset of DIO mice then underwent surgical or dietary weight loss interventions. Surgically treated DIO mice received the VSG bariatric procedure, wherein the lateral 80% of the stomach was removed, and the remaining stomach was sutured creating a tubular gastric sleeve (*Yin et al., 2012*). VSG induced a significant and sustained weight loss of 20%

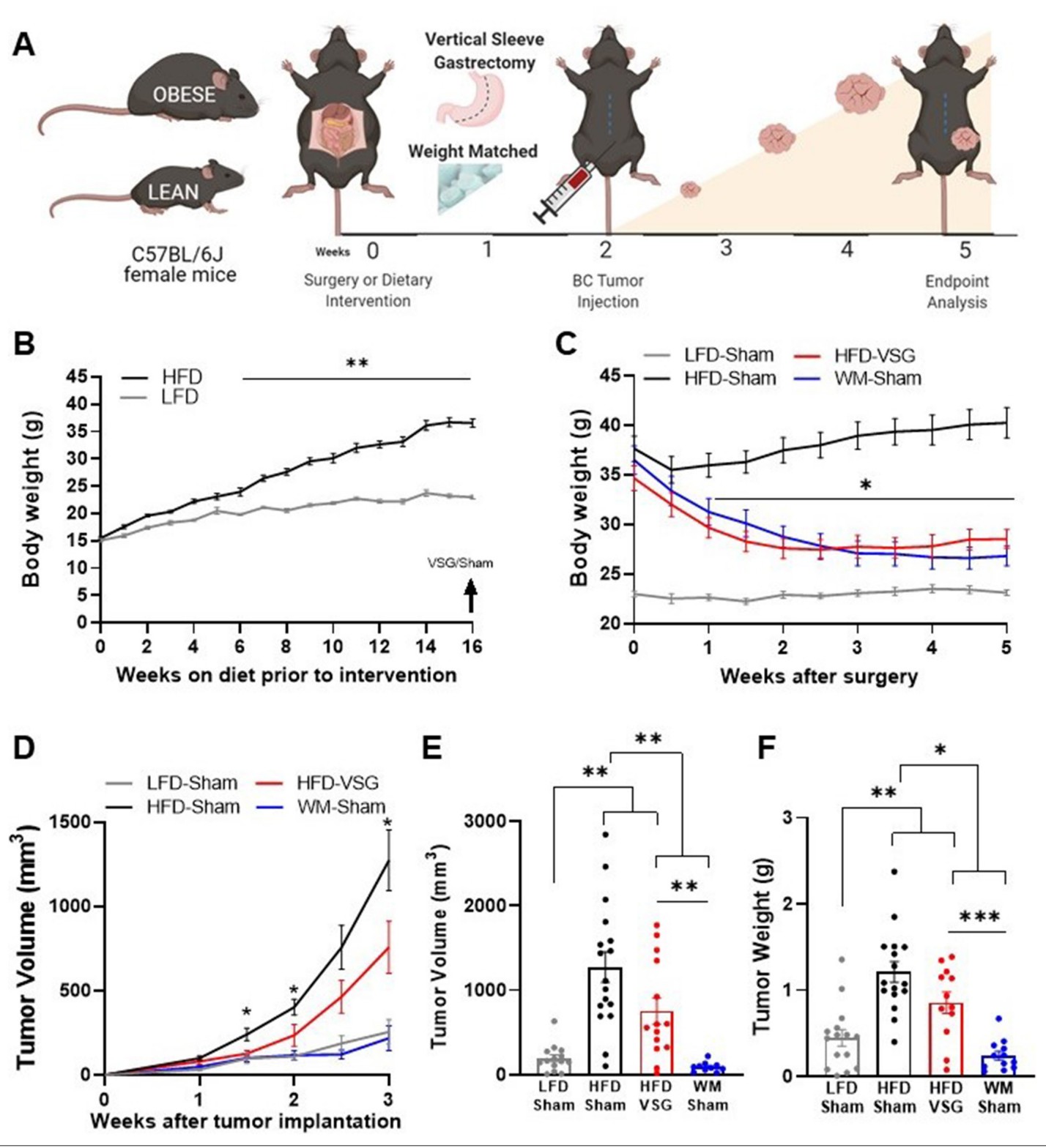

**Figure 1.** Surgical and dietary weight loss interventions reduced tumor progression and burden compared to obese mice. (**A**) Schematic of diet-induced obesity, weight loss intervention, and breast cancer cell injection in female C57BL/6J mice. Mice were fed obesogenic diets or kept lean for 16 weeks. At 20 weeks of age, mice were subjected to bariatric surgery or dietary intervention and sham surgery to stably reduce weights, while control high fat diet (HFD) and low fat diet (LFD) fed mice received sham surgery to remain obese or lean, respectively. E0771 breast cancer cells were injected at 22 weeks of age when weight loss stabilized. Tumor progression was quantified, and mice were sacrificed at endpoint 3 weeks later. (**B**) Weekly body weights

*Figure 1 continued on next page*

*Figure 1 continued*

are shown as diet-induced obesity (DIO) is established over 16 weeks on HFD compared to lean control mice fed LFD (n=15). (**C**) Body weights were measured biweekly after DIO mice were subjected to either bariatric surgery or dietary weight loss interventions. Four groups include: HFD-fed and vertical sleeve gastrectomy (HFD-VSG, red) and weight-matched (WM) caloric restricted HFD-fed and sham (WM-Sham, blue) to mirror weight loss in VSG group. These interventions were compared to controls continuously HFD-fed and sham (HFD-Sham, black) or continuously LFD-fed and sham (LFD-Sham, gray). (**D**) Tumor volume quantified over 3 weeks. (**C–D**) Two-way ANOVA Fisher's LSD test for individual comparisons with *p<0.05, and **p<0.01 signifying HFD-Sham compared to all other groups and detailed in *Supplementary file 1a and b*, respectively. (**E**) Tumor volume and (**F**) tumor weight at endpoint. (**E–F**) Mean ± SEM one-way ANOVA with Fisher's LSD test. (**B–F**) n=15 LFD-Sham, n=17 HFD-Sham, n=14 HFD-VSG, and n=13 WM-Sham. Mean ± SEM *p<0.05, **p<0.01, and ***p<0.001.

of the starting body weight, despite being continuously maintained on HFD (HFD-VSG, *Figure 1C*, detailed statistical comparisons within *Supplementary file 1a*). HFD-VSG mice lost weight to within a few grams of lean LFD-sham treated control mice. Importantly, mice did not regain weight after the VSG. Weight rebound has often been recorded in other studies in this time course (*Arble et al., 2015*; *Yin et al., 2011*). To control for the effects of surgery, all other groups that did not undergo a VSG received a sham surgery including perioperative procedures, abdominal laparotomy, anesthesia, and analgesics with minimal impacts on weight maintenance (*Figure 1A and C*). To compare the impact of VSG on breast cancer outcomes to weight loss *per se*, we employed a dietary weight loss intervention initiated after sham surgery wherein mice were fed calorically restricted amounts of HFD to match the weight loss and diet exposure of HFD-VSG treated mice, termed WM-Sham. As designed, WM-Sham body weight loss was not significantly different from HFD-VSG (*Figure 1C*). By endpoint, 5 weeks after surgical and diet interventions, both weight loss groups (HFD-VSG and WM-Sham) displayed significantly reduced body weights compared to HFD-Sham obese control mice (*Figure 1C*). These results demonstrate successful generation of complementary weight loss approaches to next investigate the impacts of bariatric surgery-mediated weight loss on tumor progression.

## Obesity-accelerated breast cancer progression was reversed by VSG and dietary weight loss

To determine if surgical weight loss corrected obesity-associated breast cancer progression, E0771 syngeneic breast cancer cells were orthotopically implanted into the fourth mammary fat pad 2 weeks following weight loss interventions, when weight loss was stabilized (*Figure 1A and C*). Tumor progression was quantified over 3 weeks (*Figure 1A and D*, detailed statistics within *Supplementary file 1b*). Breast cancer cell implantation and progression did not adversely impact body weight (*Figure 1C*). HFD-Sham tumors were significantly larger than LFD-Sham by 1 week after cell implantation. In mice that had lost weight, reduced tumor progression was observed compared to HFD-Sham from 1.5 weeks after implantation (*Figure 1D*). At endpoint, tumors were measured by caliper, then excised to quantify tumor mass. HFD-VSG tumors were significantly smaller than HFD-Sham by volume and weight (*Figure 1D–F*). However, tumors in the WM-Sham group were significantly smaller than HFD-VSG, despite identical body weights between the two weight loss approaches (*Figure 1C–F*). In fact, tumor progression was blunted in WM-Sham controls such that at endpoint, tumors in WM-Sham were not significantly different from tumors in LFD-Sham lean controls by volume or weight (*Figure 1D–F*). Thus, dietary intervention in formerly obese mice was most impactful to restore a lean-like tumor phenotype with minimal tumor progression evident and the smallest tumor burden, while weight loss by VSG proved to be less impactful to blunt tumor progression compared to weight-matched controls.

## Adiposity and leptin were reduced in formerly obese mice

Increased adiposity is associated with obesity-worsened breast cancer (*Houghton et al., 2021*). Surgical and dietary interventions resulted in a significant reduction in adiposity compared to HFD-Sham obese control mice as early as week 1 post-surgery that stabilized 2 weeks after intervention and persisted until endpoint (*Figure 2A*). Breast cancer cell implantation and progression from weeks 2 to 5 did not impact adiposity in any group (*Figure 2A*). In line with adiposity, HFD-Sham mice had about 10-fold greater mammary fat pad and gonadal adipose mass compared to lean LFD-Sham controls (*Figure 2B–C*). HFD-VSG and WM-Sham groups lost significant adipose mass compared to HFD-Sham obese controls but not to the extent quantified in lean LFD-Sham mice (*Figure 2A–C*).

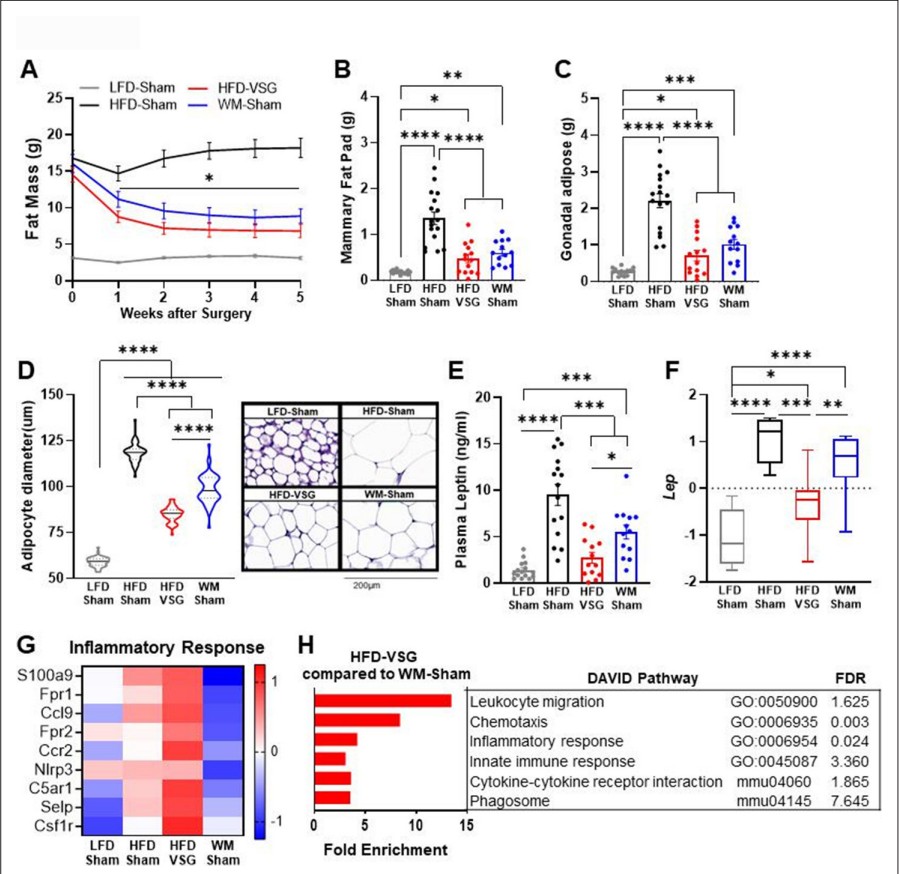

**Figure 2.** Bariatric surgery reduced adiposity similarly to weight-matched controls yet increased inflammation in mammary fat pad. (**A**) Fat mass was measured by EchoMRI. Mean ± SEM is shown. Two-way ANOVA with Fisher's LSD test, *p<0.05 all other groups compared to high fat diet (HFD)-Sham. (**B**) Mammary fat pad and (**C**) gonadal adipose weights were measured at endpoint. (**A–C**) Mean ± SEM is shown. n=15 low fat diet (LFD)-Sham, n=17 HFD-Sham, n=14 HFD-vertical sleeve gastrectomy (VSG), and n=13 weight-matched sham (WM-Sham). (**D**) Adipocyte diameter along the longest length was measured in hematoxylin and eosin sections of uninjected contralateral mammary fat pad. Violin plot with median (solid line) and quartiles (dashed line) is shown. Representative images at 20× are shown with 200 μm represented by scale bar. N=5–7, n=50 adipocytes/sample. (**E**) Circulating leptin concentration in plasma was measured at endpoint after 4 hr of fasting by Luminex assay. N=13–15. (**F**) Row mean centered gene expression of *Lep* encoding for Leptin in uninjected contralateral mammary fat pad was quantified by RNA sequencing (RNA-seq). Box and whiskers shown mean, min, and max. N=6–8. (**B–E**). One-way ANOVA with Fisher's LSD test. *p<0.05, **p<0.01, ***p<0.001, and ****p<0.0001. (**G**) Database for annotation, visualization, and integrated discovery (DAVID) analysis of regulated inflammatory pathways in mammary fat pads of HFD-VSG mice compared to WM-Sham mice. FDR: false discovery rate. (**H**) Heat map of row mean centered gene expression in uninjected contralateral mammary fat pad by RNA-seq of genes contributing to the significantly regulated Inflammatory response pathway (GO:0006954) determined by DAVID analysis. N=6–8.

Enlarged adipocyte size in the mammary fat pad is a mediator of obesity-associated inflammation and impacts breast cancer progression (*Laforest et al., 2021*). Adipocyte size in the mammary fat pad was enlarged in HFD-Sham compared to LFD-Sham mice (*Figure 2D*). HFD-VSG mammary fat pads contained significantly smaller adipocytes compared to HFD-Sham but did not reduce size to that of LFD-Sham (*Figure 2D*). Interestingly, WM-Sham mice retained significantly larger adipocytes compared to HFD-VSG, despite similar loss of adiposity and identical mammary fat pad and gonadal adipose depot weights (*Figure 2A–D*). Therefore, the association with greater adipocyte size and larger tumor burden did not hold true in these models of formerly obese mice.

Leptin is associated with adiposity and adipocyte size and can signal to activate breast cancer cell proliferation (*Lengyel et al., 2018*). Plasma leptin concentrations (*Figure 2E*) and leptin mRNA expression in the mammary fat pad (*Figure 2F*) paralleled findings for endpoint adipocyte size

(*Figure 2D*), with HFD-Sham displaying the greatest leptin plasma concentrations and mammary fat pad expression. HFD-VSG reduced leptin concentrations in plasma and in adipose tissue compared to HFD-Sham obese controls (*Figure 2E–F*). As in adipocyte size, despite comparable weight loss and adipose mass between VSG and WM-Sham groups, WM-Sham had twofold greater leptin concentration in plasma or expression in mammary fat pad compared to HFD-VSG (*Figure 2E–F*). Thus, leptin-mediated signaling does not account for why VSG is less effective in reducing tumor burden compared to weight loss alone.

## Elevated inflammation was evident in mammary fat pad uniquely after VSG weight loss intervention

Increased inflammation in the adipose has been reported in mouse models of VSG, with persistent elevations in adipose tissue macrophages despite improvements in obesity-associated parameters (*Griffin et al., 2019*; *Frikke-Schmidt et al., 2017*; *Ahn et al., 2021*; *Poitou et al., 2015*; *Lengyel et al., 2018*). Thus, we investigated if inflammatory changes in the mammary fat pad reflect pathways that could impact tumor burden using RNA sequencing (RNA-seq) analysis, database for annotation, visualization, and integrated discovery (DAVID) pathway analysis, and gene set enrichment analysis (GSEA). Compared to WM-Sham controls, HFD-VSG mammary fat pads reflected 5–10-fold elevation of immune pathways, such as leukocyte migration, chemotaxis, and inflammatory response, among others (*Figure 2G*). Examining key genes common to the inflammatory response pathways, compared to LFD-Sham lean controls, HFD-Sham obese mice displayed elevated expression of many inflammatory genes such as chemokine receptor *Ccr2* and growth factor receptor *Csf1*r, among others, as expected with DIO (*Figure 2H*). Despite significant reductions in adiposity and adipocyte size after VSG, mammary fat pads from HFD-VSG mice displayed evidence of persistent or exacerbated inflammation compared to all groups including HFD-Sham obese controls (*Figure 2H*). In stark contrast, compared to both HFD-Sham and HFD-VSG groups, mammary fat pads from WM-Sham treated mice displayed greatly reduced inflammatory gene expression to levels similar to, or lower than, lean LFD-Sham controls (*Figure 2H*). Taken together, the increased inflammatory response signature in the mammary fat pads of HFD-VSG mice suggests the possibility of a more tumor permissive environment, particularly compared to WM-Sham controls.

## Tumors displayed elevated inflammation and immune checkpoint ligand expression in mice receiving VSG

Like the mammary fat pad, transcriptome analysis of tumors in mice after VSG intervention displayed increased enrichment of inflammatory response as well as response to hypoxia pathways compared to HFD-Sham tumors, indicating an inflamed and hypoxic tumor microenvironment (*Figure 3A*), whereas these pathways were downregulated in tumors from WM-Sham mice (*Figure 3A*). Elevated pathways in VSG tumors (*Figure 3A*) contain genes - specifically *Tlr2*, *Tlr13*, *Ifngr1*, *Ccl9*, *Hif1a*, and *Cybb* - that are established to increase immune checkpoint ligand PD-L1 expression (*Figure 3B*; *Noman et al., 2014*; *Yi et al., 2021*). Therefore, we next queried immune checkpoint expression in the tumor microenvironment to determine if elevated pathways and genes in the VSG-treated group could lead to increased immune checkpoint ligand expression. Indeed, flow cytometry analysis revealed that the frequency of PD-L1+ cells was significantly and uniquely elevated in tumors after VSG intervention compared to all other groups in the CD45− fraction (*Figure 3C*). The CD45− fraction contains tumor cells as well as other stromal cells such as fibroblasts, endothelial cells, adipose stromal cells, etc. Furthermore, expression of PD-L1 quantified by mean fluorescent intensity (MFI) was also significantly elevated in the CD45− fraction from HFD-VSG tumors (*Figure 3D*). In contrast, WM-Sham intervention significantly reduced frequency of PD-L1+ non-immune cells and PD-L1 MFI relative to tumors from HFD-VSG treated mice by 60 and 30%, respectively (*Figure 3C–D*). Pro-inflammatory cytokines are associated with elevated PD-L1 through increased protein stability (*Yi et al., 2021*; *Chan et al., 2019*; *Li et al., 2020b*; *Lim et al., 2016*). Therefore, we examined circulating IL-6 using Luminex. Compared to HFD-Sham, circulating IL-6 was significantly elevated in HFD-VSG (*Figure 3E*). In contrast, WM-Sham mice displayed a 3.3-fold significantly reduced concentration of IL-6 compared to mice in the HFD-VSG group (*Figure 3E*). In E0771 breast cancer cells, treatment with IL-6 increased PD-L1 MFI as quantified by flow cytometry. Similarly, GSEA revealed significant enrichment of the hallmark IL-6/Jak/STAT3 signaling pathway in tumors from HFD-VSG group compared to WM-Sham

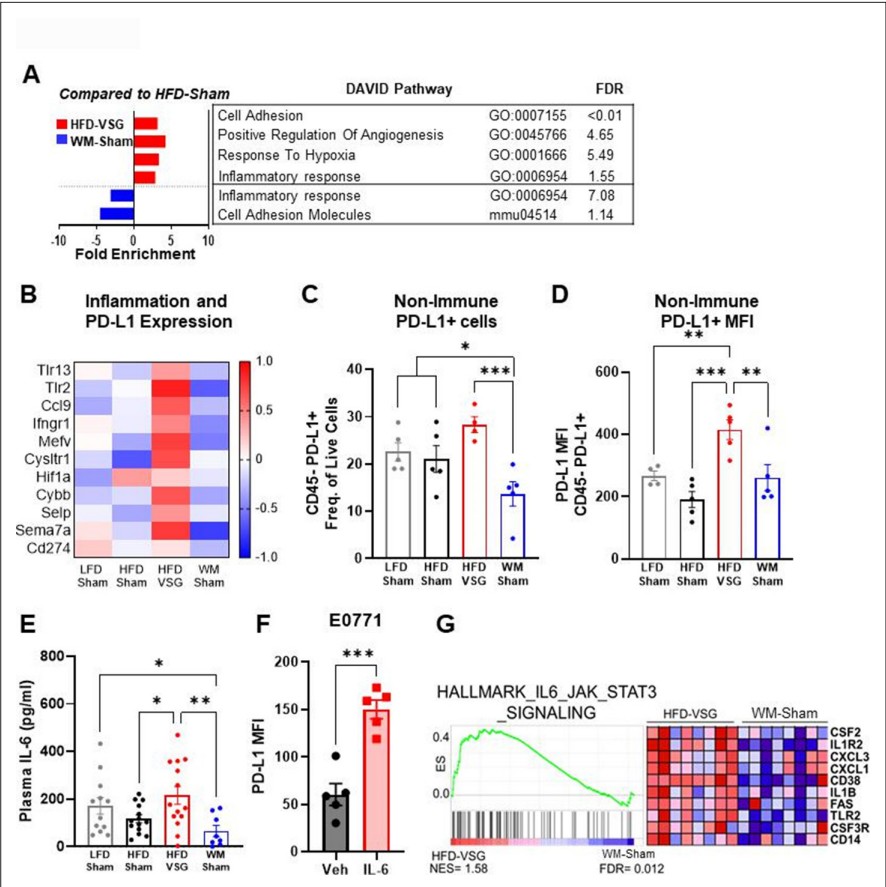

**Figure 3.** The tumor microenvironment displayed increased inflammation and immune checkpoint ligand expression following bariatric surgery. (**A**) Database for annotation, visualization, and integrated discovery (DAVID) analysis of regulated pathways and false discovery rate (FDR) for high fat diet (HFD)-vertical sleeve gastrectomy (VSG) (red) and weight-matched sham (WM-Sham) (blue) relative to tumors from HFD-Sham mice is shown. N=6–8. (**B**) Heat map of row mean centered gene expression in tumor by RNA sequencing (RNA-seq) of genes contributing to significantly regulated inflammatory response pathway (GO:0006954) and response to hypoxia pathway (GO:0001666) determined by DAVID analysis. N=6–8. (**C**) Flow cytometric analysis of CD45 negative (CD45−) PD-L1+ non-immune cells in tumor is plotted as frequency of total live cells. (**D**) Mean fluorescent intensity (MFI) of PD-L1 on CD45− PD-L1+ cells in tumor is shown. N=4–5. (**E**) Circulating IL-6 concentration in plasma was measured at endpoint after 4 hr of fasting by Luminex. N=8–14. (**F**) Flow cytometric analysis of PD-L1 MFI in E0771 breast cancer cells after treatment with recombinant mouse IL-6 (200 pg/mL) for 4 hr. Mean ± SEM is shown. One-way ANOVA with Fisher's LSD test. *p<0.05, **p<0.01, and ***p<0.001. (**G**) Gene set enrichment analysis (GSEA) of the hallmark pathway for IL6/JAK/STAT3 gene set from the Molecular Signatures Database of the Broad Institute is reported in HFD-VSG tumors compared to WM-Sham controls. The normalized enrichment score (NES) and FDR are shown.

tumors (*Figure 3G*). Overall, surgically induced weight loss increased tumor cell specific and circulating inflammation and elevated the immune checkpoint ligand PD-L1 in the tumor microenvironment suggesting the presence of impaired anti-tumor immunity (*Ngiow and Young, 2020*; *Muenst et al., 2014*).

## T cell tumor content and cytolysis were impaired after VSG

In the tumor microenvironment, high PD-L1 expression by tumor cells can dampen T cell-mediated anti-tumor immune responses (*Yi et al., 2021*; *Ngiow and Young, 2020*; *Muenst et al., 2014*). Therefore, we next investigated T cell content and associated activation pathways by flow cytometry and RNA-seq (*Wang et al., 2019*). CD3+ T cell frequency in tumors from HFD-VSG mice was significantly decreased compared to tumors from LFD-Sham control mice (*Figure 4A*). In contrast, CD3+ T cell frequency in weight-matched controls was significantly greater compared to content in tumors

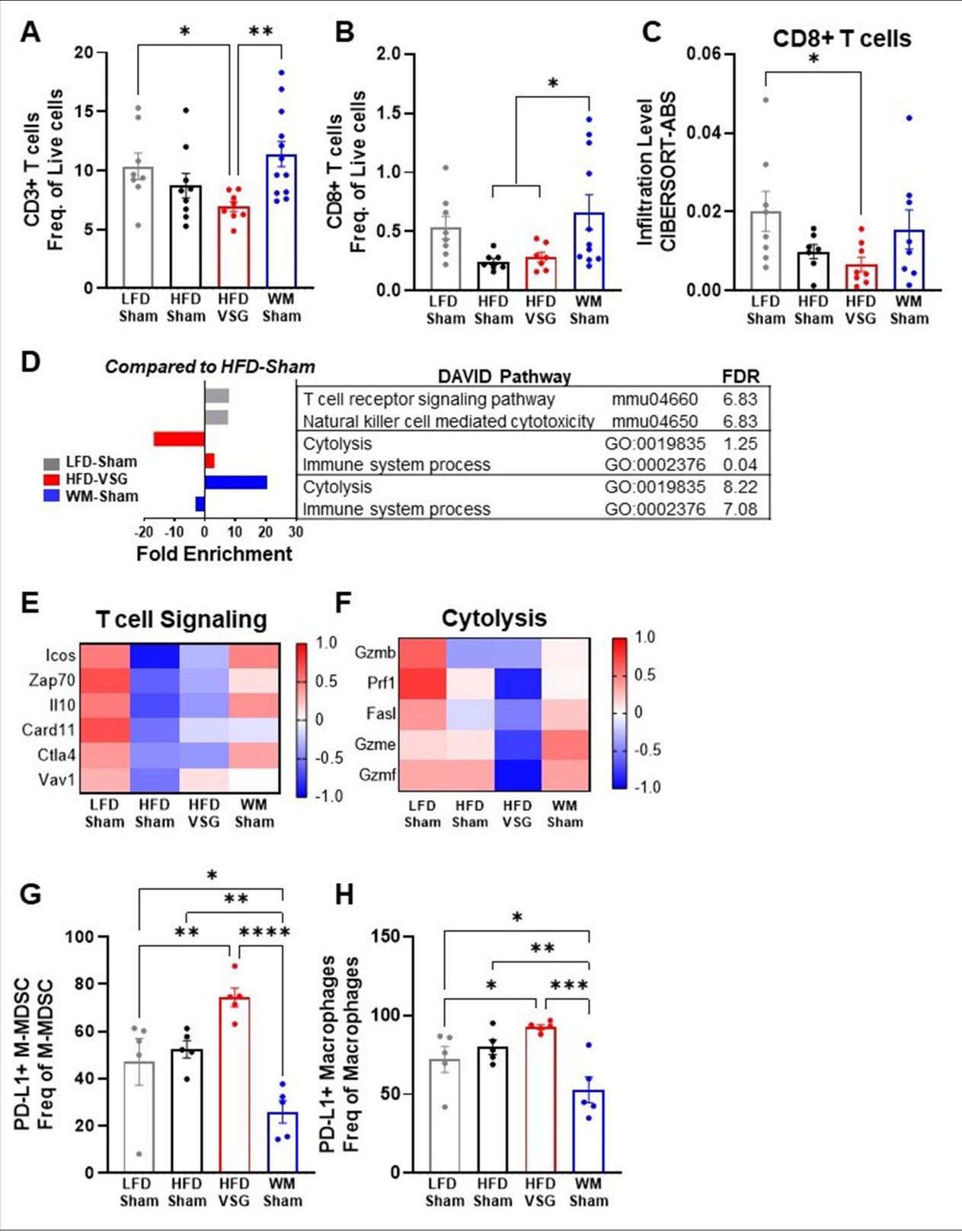

**Figure 4.** Vertical sleeve gastrectomy (VSG) reduced CD8+ tumor T lymphocyte frequency and markers of T cell activation demonstrating impaired anti-tumor immunity. (**A–B**) Flow cytometric analysis of tumor (**A**) CD3+ T cells and (**B**) CD8+ T cells is shown as frequency of total live cells. N=8–12. (**C**) Analysis of tumor CD8+ T cell content from RNA sequencing (RNA-seq) data using the cell-type identification estimating relative subsets of RNA transcripts (CIBERSORT)-Abs algorithm in TIMER2.0. N=6–8. (**D**) Database for annotation, visualization, and integrated discovery (DAVID) analysis of

*Figure 4 continued on next page*

Figure 4 continued

regulated pathways for low fat diet (LFD)-Sham (gray), high fat diet (HFD)-VSG (red), and weight-matched sham (WM-Sham) (blue) relative to tumors from HFD-Sham mice. N=6–8. (**E**) Heat map of row mean centered gene expression in tumor by RNA-seq of genes contributing to the significantly regulated T cell signaling pathway (mmu04660 and false discovery rate [FDR] 6.83) and (**F**) cytolysis (GO:0019835 and FDR 1.25) as determined by DAVID analysis. N=6–8. (**G**) Flow cytometric analysis of tumor PD-L1+ monocytic myeloid derived suppressor cells (M-MDSC) shown as frequency of total M-MDSC. N=5. (**H**) Flow cytometric analysis of tumor PD-L1+ macrophages shown as frequency of total macrophages. N=5. (**A–C** and **G–H**) Mean ± SEM are shown. One-way ANOVA with Fisher's LSD test *p<0.05, **p<0.01, ***p<0.001, and ****p<0.0001.

The online version of this article includes the following figure supplement(s) for figure 4:

**Figure supplement 1.** CD3+ and CD8+ T cell frequencies and CD3+ PD-1 expression by MFI were unchanged in tumor draining lymph node (TdLN) and tumors.

after VSG (*Figure 4A*). Obesity has been shown to decrease CD8+ cytotoxic tumor T cells (*Wang et al., 2019*; *Pingili et al., 2021*) which was evident, but not significant, in this study comparing lean LFD-Sham to obese HFD-Sham controls (*Figure 4B–C*). Obesity-driven CD8+ T cell reductions were not corrected in tumors from formerly obese HFD-VSG mice by both flow and RNA-seq cell-type identification estimating relative subsets of RNA transcripts (CIBERSORT) analysis using TIMER2.0 (*Figure 4B–C*). Importantly, obesity-driven reductions in CD8+ T cell frequencies were reversed in tumors from WM-Sham control mice and corrected to levels found in tumors from lean LFD-Sham controls (*Figure 4B–C*). Transcriptomic analysis revealed that T cell specific signaling pathways and genes in the tumor mirrored T cell content (*Figure 4D–E*). Lowest T cell signaling gene signature expression was evident in tumors from HFD-Sham and HFD-VSG mice, with some correction in WM-Sham mice toward levels detected in lean LFD-Sham controls (*Figure 4D–E*). Of note, CD3+ and CD8+ T cell frequencies were unchanged in the tumor adjacent mammary fat pad and tumor draining lymph node (TdLN) (*Figure 4—figure supplement 1A-B*), suggesting T cell changes were specific to the tumor microenvironment. Furthermore, neither T cells in tumor nor TdLN displayed changes in PD-1 expression measured by MFI (*Figure 4—figure supplement 1C-D*).

A critical function of anti-tumor immune cells is effective cytolytic activity (*Wang et al., 2019*). RNA-seq analysis showed that the cytolysis pathway was significantly and potently downregulated by 17-fold in HFD-VSG tumors compared to obese HFD-Sham controls (*Figure 4D*). In contrast, tumors from the WM-Sham intervention group displayed the greatest activation with over 20-fold increase in the cytolysis pathway (*Figure 4D*). Genes in the cytolytic pathway were greatly downregulated in HFD-VSG tumors compared to all other groups including granzymes and fas ligand (*Gzmb*, *Prf1*, *Fasl*, *Gzme*, and *Gzmf*), while gene expression was reversed to lean-like levels in tumors from WM-Sham mice (*Figure 4F*).

To investigate potential mechanisms known to impact T cell signaling and activation such as elevated cytolysis markers including granzymes, we next examined immune cells that impair T cell activation by flow cytometric analysis. HFD-VSG tumors displayed elevated PD-L1+ monocytic myeloid derived suppressor cells (M-MDSC, *Figure 4G*) and macrophages (*Figure 4H*) relative to all other diet and surgical groups. Compared to HFD-VSG tumors, M-MDSC displayed a significant 2.9-fold reduction in tumors in the WM-Sham group. Similarly, compared to HFD-VSG tumors, PD-L1+ macrophages displayed a significant 1.76-fold reduction in tumors in the WM-Sham group (*Figure 4G–H*, respectively). PD-L1+ is a marker of immunosuppressive capacity. PD-L1+ M-MDSCs and macrophages would impair T cell activation by inducing apoptosis or exhaustion (*Crespo et al., 2013*; *Adeshakin et al., 2022*; *Hou et al., 2020*). Taken together, weight-matched control mice displayed uniquely restored T cell content and signaling pathways that were depressed by obesity which suggests an apparent effective anti-tumor response aligning with reduced tumor burden. Plus, PD-L1+ cells associated with immunosuppressive capacity were greatly reduced in WM-Sham tumors. In contrast, mice after VSG displayed a tumor microenvironment that resembled persistent obesity or elevated presence of PD-L1+ immunosuppressive MDSCs and macrophages, with reduced T cell content and cytolytic markers, despite comparable weight loss with weight-matched controls.

## Anti-PD-L1 therapy was more efficacious in VSG mice

The elevation of tumor immune checkpoint ligand PD-L1 after bariatric surgery may be one mechanism that underlies why surgical weight loss was less effective in reducing obesity-worsened tumor growth compared to weight loss alone. Therefore, we hypothesized that ICB would re-invigorate

the anti-tumor immune response in mice after VSG to reduce tumor burden. Higher expression of PD-L1 in tumors is associated with longer overall survival in patients treated with ICB (*Liu et al., 2020*). Mice were weaned onto diets and received surgical or dietary weight loss interventions prior to tumor engraftment as above (*Figure 1A*). Mice were then treated with anti-PD-L1 or isotype control IgG2b. Anti-PD-L1 did not affect body weight, mammary fat pad, or gonadal adipose weight, suggesting no negative impacts on systemic homeostasis (*Figure 5—figure supplement 1*). In LFD-Sham lean controls, despite the tumor being sixfold smaller than in obese mice at baseline, anti-PD-L1 significantly reduced tumor growth over time (*Figure 5A*). HFD-Sham mice were completely resistant to ICB (*Figure 5A–B*). Notably, anti-PD-L1 significantly reduced tumor progression in HFD-VSG (*Figure 5A*), with significantly reduced tumor volume at endpoint (*Figure 5B*). In line with an already active anti-tumor immune response, ICB was moderately and insignificantly effective in WM-Sham mice (*Figure 5A–B*). Thus, ICB was efficacious in reducing tumor progression in mice after HFD-VSG to sizes comparable to tumors found in lean mice.

ICB restores cytotoxic T cell function, thus re-establishing effective anti-tumor immunity (*Ngiow and Young, 2020*). While there were not significant differences in mean CD8+ T cell content at endpoint (*Figure 5C*), evidence of cytolytic capacity is upregulated in VSG tumors treated with anti-PD-L1 with increased *Ifng*, *Gzmb*, and *Prf1* expression (*Figure 5D–F*). Our results suggest that ICB compensates for an ineffective anti-tumor immunity associated with elevated PD-L1 expression in the tumors of VSG mice to restore markers of cytotoxic T cell response, which leads to reduced tumor burden.

## A BSAS derived from patient and murine adipose tissue associates with tumor burden

To determine if genes associated with weight loss after bariatric surgery are conserved across species, we compared subcutaneous adipose tissue biopsies from female human subject samples before and after bariatric surgery using a publicly available dataset (*Poitou et al., 2015*) with mammary fat pad tissue isolated from HFD-Sham and HFD-VSG mice in study 1 above (*Figure 6A*). When comparing transcriptomic changes in adipose tissue after bariatric surgery from both humans and mouse models, there were 54 differentially expressed genes (DEGs) in common (*Figure 6A*), which we termed the BSAS (*Supplementary file 1c*). Overlapping DEGs identified pathways involved in metabolism and adipose tissue remodeling after weight loss and immune system processes by DAVID pathway analysis (*Figure 6B*). We next examined the relationship between BSAS and tumor burden in our models with divergent tumor growth patterns. Of the 54 genes in this BSAS, 11 genes significantly correlated to volumes of HFD-Sham and HFD-VSG tumors, which is shown in *Figure 6C*. We termed these 11 genes the tumor associated BSAS (T-BSAS) gene signature (*Figure 6C*). Seven of the genes were downregulated by obesity and reversed by VSG specific weight loss including *Ido1*, *Aldoc*, *Tmem125*, *Dgki*, *Slc7a4*, *Msc*, *and Ephb3*, while four were inversely regulated with obesity elevating *Klhl5*, *Nek6*, *Arhgap20*, and *Hp*. For example, *Ido1* expression relative to tumor size shows a significant negative correlation (*Figure 6D*). Overall, compared to the HFD-Sham obese group, the T-BSAS signature in HFD-VSG tumor largely resembled tumors from LFD-Sham (*Figure 6C*). This multi-species approach uniquely demonstrates conserved transcriptional responses impacted by bariatric surgery that associates with tumor burden.

## Discussion

Obesity was identified as a cancer risk factor almost 20 years ago, with 13 obesity-associated cancers now recognized (*Calle et al., 2003*; *Lauby-Secretan et al., 2016*). Obesity negatively impacts many cancer outcomes and is thus a potential modifiable factor (*Bohm et al., 2022*; *Bhardwaj and Brown, 2021*). Murine models examining weight loss through diet switch, caloric restriction, or time-restricted feeding (fasting) support that weight loss impairs tumor progression (*Das et al., 2021*; *Lv et al., 2014*; *Hursting et al., 2003*; *Qin et al., 2016*). However, dietary weight loss alone is minimally effective for patients and difficult to maintain. The use of bariatric surgical approaches to induce durable weight loss is increasing in prevalence. In this study, to investigate the impacts of weight loss by bariatric surgery on subsequent tumor burden, we first established a murine model wherein once weight loss is stabilized, cancer cells were orthotopically implanted to examine progression and burden. We show that tumor growth in formerly obese mice that lost weight through either bariatric

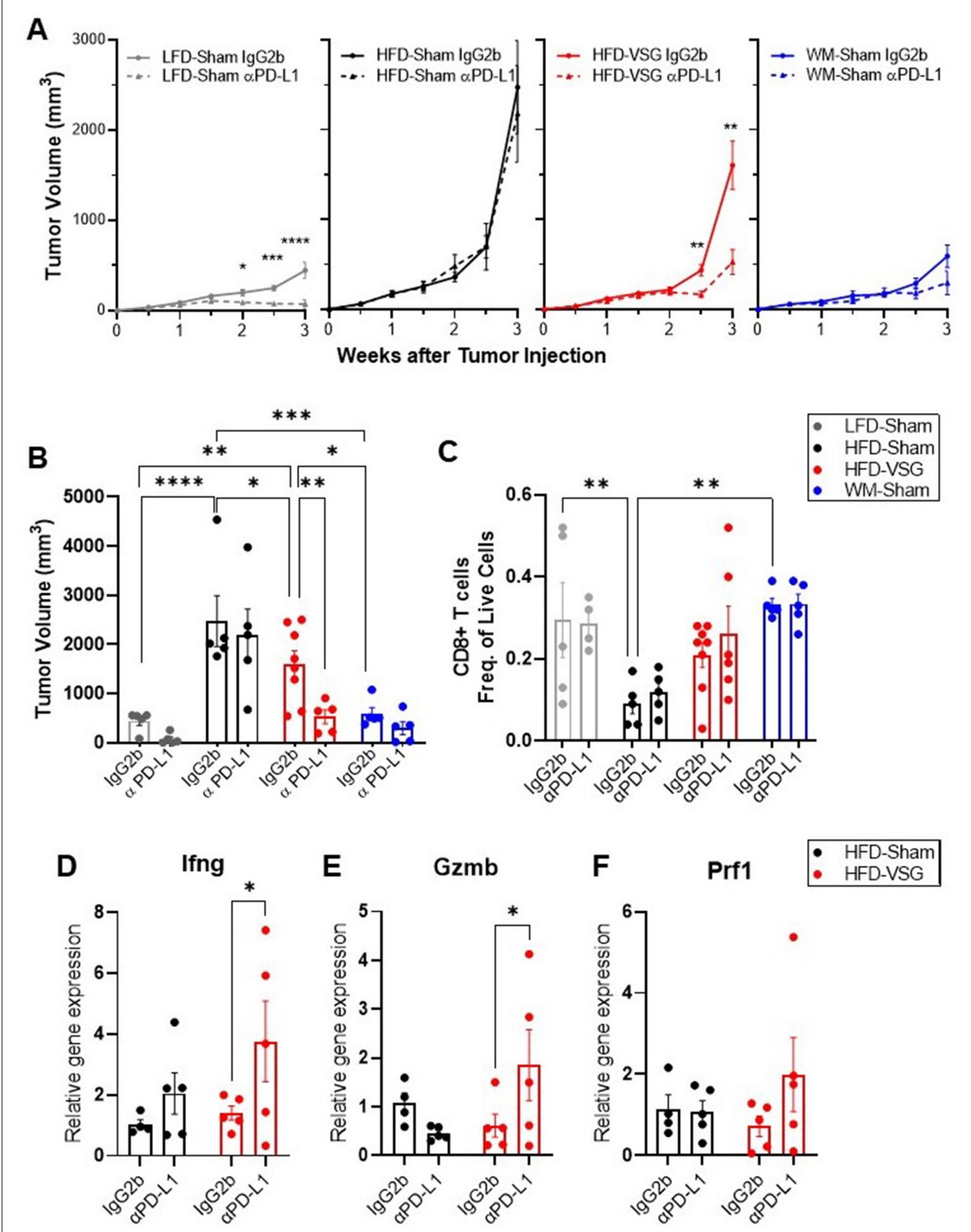

**Figure 5.** Immune checkpoint blockade re-invigorated the anti-tumor immune response in mice after bariatric surgery. Diet-induced obesity (DIO) mice were subjected to either surgical or dietary weight loss interventions and compared to lean or obese controls similar to *Figure 1A*. After weight stabilization at 2 weeks, mice were injected with E0771 cells, as above. Mice were either treated with anti-PD-L1 or IgG2b isotype control every 3 days until sacrifice at 3 weeks after cell injection. (**A**) Mean tumor growth in each diet group treated with anti-PD-L1 or IgG2b isotype control is shown.

*Figure 5 continued on next page*

*Figure 5 continued*

(**B**) Tumor volume at endpoint. (**C**) Flow cytometric analysis of CD8+ T cells as frequency of total live cells in tumor. (**D**) Relative gene expression normalized to 18S of *Ifng* (**E**), *Gzmb,* and (**F**) *Prf1* in tumors. (**A–F**) Mean ± SEM. N=5–8. Two-way ANOVA with Fisher's LSD test. Only relevant statistical comparisons are shown for clarity. *p<0.05, **p<0.01, ***p<0.001, and ****p<0.0001.

The online version of this article includes the following figure supplement(s) for figure 5:

**Figure supplement 1.** Immune checkpoint blockade did not alter body weight or adiposity.

surgical intervention with VSG or weight-matched controls were effective at blunting breast cancer progression and reducing tumor burden. Thus, in mice from the VSG group and weight-matched control groups, results suggest that tumor responses aligned with adiposity not diet exposure. Both groups were fed the same HFD as obese mice which presented with the greatest adiposity and largest tumors, suggesting that diet *per se* is not as important as adiposity in driving tumor progression. However, bariatric surgery only partially reduced obesity accelerated breast cancer progression while weight-matched controls effectively blunted growth to a lean-like phenotype.

Some mechanisms linking obesity-driven breast cancer include elevated adipokines, chronic inflammation, and dampened anti-tumor immune response (*Lengyel et al., 2018*; *Naik et al., 2019*). We examined multiple factors associated with obesity and metabolic dysfunction, including extent of weight loss, adiposity, mammary fat pad adipocyte size, and local or circulating leptin levels; none were associated with changes in tumor burden in formerly obese mice. However, RNA-seq analysis of

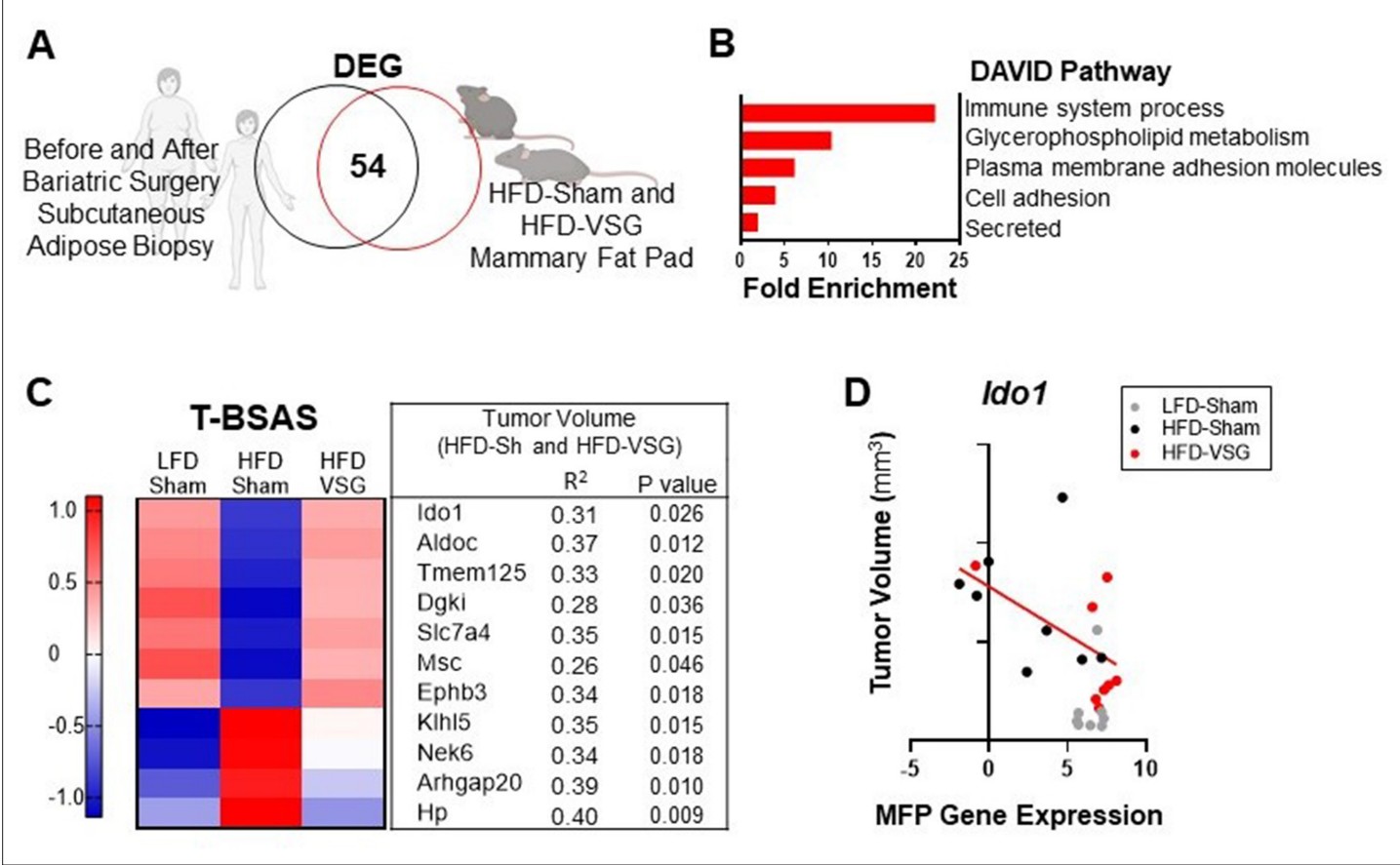

**Figure 6.** Conserved adipose bariatric surgery-associated weight loss signature associated with tumor volume. (**A**) Venn diagram of differentially expressed genes (DEGs) from obese and lean patient subcutaneous adipose tissue before and 3 months after bariatric surgery, respectively, compared to obese high fat diet (HFD)-Sham and lean HFD-vertical sleeve gastrectomy (VSG) mammary fat pad. (**B**) Database for annotation, visualization, and integrated discovery (DAVID) pathways enriched in the overlapping DEG are indicated. (**C**) A tumor bariatric surgery-associated weight loss signature (T-BSAS) signature was identified as a subset of BSAS genes that significantly correlated to tumor volume. Heat map of row mean centered expression of T-BSAS genes in the mammary fat pad by RNA sequencing (RNA-seq). (**D**) Tumor volume compared to unaffected mammary fat pad (MFP) gene expression of *Ido1* is plotted. Simple linear regression (red line) for HFD-Sham and HFD-VSG groups is shown (R²=0.31 and p=0.026).

the tumor and mammary fat pad demonstrated critical inflammatory pathways regulated by obesity and weight loss. Despite a significant reduction in tumor burden compared to obese HFD-Sham mice, VSG-treated mice demonstrated upregulated mammary fat pad inflammation to levels greater than those of obese mice. Our finding of elevated inflammation in the mammary fat pad after VSG is consistent with several studies reporting inflammation in adipose depots following bariatric surgery in murine models (*Griffin et al., 2019*; *Frikke-Schmidt et al., 2017*; *Ahn et al., 2021*; *Poitou et al., 2015*; *Harris et al., 2020*). The persistent inflammation identified after bariatric surgery in adipose tissue could be due to adipose remodeling following rapid weight loss or wound repair signaling from the surgical injury itself. However, these inflammatory changes to the mammary fat pad were uniquely induced by the VSG bariatric surgery, not likely due to surgery itself, since all other groups received a sham surgery as controls. In addition to the mammary fat pad, we report activation of inflammatory and hypoxic pathways in the tumors of mice after VSG but not in other interventions. Therefore, future studies to determine the extent and timing of bariatric surgery associated remodeling in both murine models and humans are warranted. While the murine model presented herein demonstrated successfully stabilized weight loss, most other reports demonstrate weight rebound within a few weeks postsurgery, which should be optimized in future cancer studies (*Arble et al., 2015*; *Yin et al., 2011*).

We posited that inflammation, including circulating and the surrounding adipose and tumor, led to dramatic elevations in PD-L1 expression on non-immune and myeloid cells detected uniquely after VSG. The CD45− fraction contains tumor cells as well as other stromal cells such as adipocytes, adipose stromal cells, mesenchymal stem cells, mast cells, etc. which have been reported to express PD-L1 (*Wu et al., 2020*; *Wu et al., 2018*; *Hirano et al., 2021*). It is likely that several cell types display elevated PD-L1 in the tumor microenvironment. PD-L1 is stabilized by pro-inflammatory cytokines such as IL-6 (*Yi et al., 2021*; *Chan et al., 2019*). Depressed CD3+ and CD8+ T cell content and dampened expression of T cell cytolytic markers detected in tumors after VSG intervention could have hindered effective anti-tumor immunity after bariatric surgery-associated weight loss. These changes in PD-L1 on non-immune and myeloid cells, and T cell content and signaling, or cytolytic pathway were not present in the weight-matched controls despite this group losing the same amount of weight as VSG intervention. In fact, weight-matched controls had significantly elevated cytotoxic T cell tumor content and evidence of cytolytic function and reduced PD-L1+ immunosuppressive M-MDSCs and macrophages which associate with reduced tumor burden. Taken together, it is likely that the elevated PD-L1+ CD45− cells after VSG, as well as PD-L1+ macrophages and M-MDSCs led to reduced T cell signaling and activation, which would reduce CD3+ and CD8+ T cell content (*Crespo et al., 2013*; *Nelson et al., 2021*).

Tumor inflammation and hypoxia increase expression of PD-L1 within the tumor microenvironment (*Yi et al., 2021*). Inflammation in the obese tumor microenviroment further exacerbates immune checkpoint expression and PD-L1+ cells thus enabling worsened outcomes (*Li et al., 2020b*; *Wang et al., 2019*; *Woodall et al., 2020*; *Cha et al., 2018*). Patient tumors with high PD-L1 expression are enriched in inflammation, cell adhesion, and angiogenesis pathways (*Qi et al., 2020*; *Billon et al., 2019*), which were pathways upregulated in tumors after VSG. Furthermore, tumors from mice that received VSG had high expression of genes that are also enriched in patient tumors that are positive for PD-L1 including *Mefv*, *Selp*, *Sema7a*, and *Cysltr1* (*Billon et al., 2019*) which are critically linked to responsiveness to ICB. Increasing evidence supports that obesity improves immunotherapy efficacy in melanoma and other cancers, and studies in breast cancer are ongoing (*Richtig et al., 2018*; *McQuade et al., 2018*; *Cortellini et al., 2019*). Here, we report for the first time that anti-PD-L1 was most effective in reducing tumor burden in the mice that received VSG to induce weight loss, with restored expression of cytolytic genes. Taken together, we have identified unique anti-tumor efficacy of anti-PD-L1 in mice after VSG.

Finally, we determined genes associated with weight loss after bariatric surgery conserved across species. We took advantage of published transcriptomes of subcutaneous adipose tissue from female patients before and after bariatric surgery in comparison with mammary fat pad expression from obese and formerly obese mice after VSG bariatric surgery. We identified a novel weight loss signature specific to bariatric surgery conserved between mice and humans, termed BSAS. Pathways associated with metabolism, remodeling, and immune cells were identified from conserved genes. Because our study consisted of surgical versus dietary interventions and cancer progression, we are in the unique position to compare BSAS transcriptomic changes to tumor outcomes,

which we termed T-BSAS. We demonstrate that a subset of 11 key genes in the T-BSAS signature was associated with tumor outcomes in our mouse models. For example, *Ido1*, indoleamine 2, 3-dioxygenase, is part of the rate limiting enzyme that metabolizes L-tryptophan to N-formylkynurenine. The conserved BSAS gene list demonstrated that compared to obese state, *Ido1* was increased by bariatric surgery in both mouse and human. Of note, *Ido1* was not elevated by WM-induced weight loss in our study (data not shown), which suggests that changes in *Ido1* expression could be a specific response to surgically induced weight loss. Over-expression of IDO depletes tryptophan, leading to accumulation of tryptophan metabolites which can induce immunosuppression. Thus, IDO plays a central role in immune escape through reduced CD8+ T cell activation and increased T cell death (*Zhai et al., 2020*) with multiple IDO inhibitors under investigation (*Tang et al., 2021*). We previously reported that *Ido1* expression in the tumor adjacent mammary fat pad was decreased after anti-PD-1 immunotherapy in obese mice (*Pingili et al., 2021*). Thus, the aberrant upregulation of IDO after bariatric surgery-induced weight loss is one potential mechanism limiting anti-tumor immunity in our VSG model that remains under investigation. One limitation of our study is that this study examines just a single syngeneic orthotopically transplanted model wherein we have examined impact of obesity and weight loss on tumor progression and response to immunotherapy. Future work will investigate other cancer models; however, few models exist to study both highly obesogenic strains and breast cancer (*Bohm et al., 2022*). Additionally, variables such as duration of obesity, extent of surgery, and time post-recovery will likely impact immune parameters and should be investigated in pre-clinical and patient settings. It is also possible that different dosing or timing of ICB or combination therapy would demonstrate a greater inhibition of tumor progression.

In patients, weight loss has been shown to improve prognosis after breast cancer has already been diagnosed (*Seiler et al., 2018*; *Chlebowski et al., 2006*; *Patterson et al., 2018*). In practice, preventing obesity or promoting weight loss has been a difficult and complex public health challenge. Important retrospective work has shown that patients who underwent bariatric surgery had reduced risk of both pre-menopausal and post-menopausal breast cancer with a 64% reduced risk in pre-menopausal ER-tumors, typically the most aggressive tumors with the worst outcomes (*Feigelson et al., 2020*). Furthermore, reduced recurrence and mortality from cancer have been observed in bariatric surgery patients (*Aminian et al., 2022*; *Bruno and Berger, 2020*; *Zhang et al., 2020a*) although underlying mechanisms remain unclear. A major question remains regarding whether reductions in cancer risk and outcomes are associated with weight loss *per se* or are due to bariatric surgery-specific benefits, which are inherently challenging to delineate in patients (*Schauer et al., 2017*; *Alvarez et al., 2018*). Taken together, additional prospective studies are necessary to determine if intentional weight loss through surgery offers significant promise as an approach that could be highly impactful for reducing cancer burden and potentially improving therapy (*Courcoulas, 2022*).

In sum, despite successful and sustained weight loss, tumors in formerly obese mice that received VSG bariatric surgery failed to display sufficiently improved anti-tumor immunity like controls that lost similar amounts of weight. Elevated inflammation in the mammary fat pad and tumor reduced cytotoxic T cells suggested an ineffective anti-tumor milieu after VSG. Anti-PD-L1 immunotherapy was able to improve tumor outcomes in surgical weight loss mice. Ultimately bariatric surgery is the most effective long-term weight loss solution and could be considered in cancer prevention for high-risk obese patients to reduce cancer risk or recurrence. Clinical trials are underway in some severely obese patients with studies examining changes in breast density and breast cancer risk after bariatric surgery (*ClinicalTrials, 2021*), reviewed by *Bohm et al., 2022*. Understanding how obesity impacts breast cancer anti-tumor immunity and determining effective weight loss strategies to maximize response to therapies will be valuable. In this study, we queried response to ICB in obese and weight loss models, but response to chemotherapy and radiation therapy and combined therapies are also important areas of investigation to advance the field. Because one-third of Americans are considered obese and 9.2% currently severely obese (*Hales, 2020*), this study is an important first step in understanding bariatric surgery impacts on cancer progression and immunotherapy.

## Materials and methods

**Key resources table**

| Reagent type (species) or resource | Designation | Source or reference | Identifiers | Additional information |
|---|---|---|---|---|
| Strain and strain background (*Mus musculus*) | C57BL/6J | The Jackson Laboratory | JAX:000664 | Female |
| Cell line (*Mus musculus*) | Breast cancer | Korkaya (*Ouzounova et al., 2017*) | E0771-luciferase | Cell purchased from ATCC and transfected with luciferase (*Ouzounova et al., 2017*) were a generous gift from Korkaya. |
| Antibody | Anti-mouse CD45 violetFluor 450 (Rat monoclonal) | Tonbo Biosciences | Cat# 75–0451 U025 | (1:40) |
| Antibody | Anti-mouse CD3ε Brilliant Violet 785 (Armenian Hamster monoclonal) | BioLegend | Cat# 100,355 | (1:40) |
| Antibody | Anti-mouse CD8a FITC (Rat monoclonal) | Tonbo Biosciences | Cat# 35–0081 U025 | (1:100) |
| Antibody | Anti-mouse CD274 Brilliant Violet 711 (Rat monoclonal) | BioLegend | Cat# 124,319 | (1:10) |
| Antibody | Anti-mouse PD-1 Brilliant Violet 421 (Rat monoclonal) | Biolegend | Cat# 135,217 | (1:10) |
| Antibody | Anti-mouse CD11b Red-Fluor 710 (Rat monoclonal) | Tonbo Biosciences | Cat# 80–0112 U025 | (1:20) |
| Antibody | Anti-mouse Ly-6C APC (Rat monoclonal) | Biolegend | Cat# 128,015 | (1:40) |
| Antibody | Anti-mouse Ly-6G PerCP-Cyanine 5.5 (Rat monoclonal) | Tonbo Biosciences | Cat# 65–1276 U025 | (1:40) |
| Antibody | Anti-mouse F4/80 PE (Rat monoclonal) | Tonbo Biosciences | Cat# 50–4801 U025 | (1:40) |
| Peptide, recombinant protein | Interleukin-6 | Shenandoah Biotechnology Inc | Cat# 200–02 | (200 pg/mL) |
| Sequence-based reagent | Ifng Primer | IDT | F:GGATGCATTCATGAGTATTGC R:GTGGACCACTCGGATGAG | |
| Sequence-based reagent | Prf1 Primer | IDT | F:GAGAAGACCTATCAGGACCA, R:AGCCTGTGGTAAGCATG, | |
| Sequence-based reagent | Gzmb Primer | IDT | F:CCTCCTGCTACTGCTGAC, R:GTCAGCACAAAGTCCTCTC | |
| Sequence-based reagent | Gzmb Primer | IDT | F:TTCGGAACTGAGGCCATGATT, R:TTTCGCTCTGGTCCGTCTTG | |
| Antibody | Anti-PD-L1 (Rat monoclonal) | BioXcell | Clone 10 F.9G2, #BE0101 | (8 mg/kg) |
| Antibody | IgG2b isotype control (Rat monoclonal) | BioXcell | Clone LTF-2, #BE0090 | (8 mg/kg) |

## Reagents

All reagents were obtained from Sigma-Aldrich (St. Louis, MO) unless otherwise noted. Fetal bovine serum (FBS, Gibco, Waltham, MA), RPMI 1640 (Corning, Tewksbury, MA), 100× L-glutamine, 100× penicillin/streptomycin HyClone (Pittsburgh, PA), and Gibco 100× antibiotic mix were obtained from

Thermo Fisher (Waltham, MA). Matrigel is from Corning (Tewksbury, MA). Antibodies for flow are described in key resources table and purchased from Tonbo (San Diego, CA), Thermo Fisher, and Biolegend (San Diego, CA).

## Mice and diets

Animal studies were performed with approval and in accordance with the guidelines of the Institutional Animal Care and Use Committee (IACUC) at the University of Tennessee Health Science Center (Animal Welfare Assurance Number A3325-01) and in accordance with the National Institutes of Health Guide for the Care and Use of Laboratory Animals. The protocol was approved under the protocol identifier 21.0224. All animals were housed in a temperature-controlled facility with a 12 hr light/dark cycle and ad libitium access to food and water, except where indicated. Three-week-old female C57BL/6J (Jackson stock number: 000664) mice were shipped to UTHSC and acclimated 1 week. Four-week-old mice were randomized to either obesogenic HFD (D12492i – 60% kcal derived from fat) or LFD (D12450Ji- 10% kcal derived from fat) from Research Diets Inc (New Brunswick, NJ) for 16 weeks (age 4 weeks to 20 weeks old, study design *Figure 1A*). Mice resistant to DIO, as defined by less than 28 g after 16 weeks of HFD, were excluded from the study. DIO mice received either a bariatric surgery or sham control surgery and dietary intervention as described below.

## Body weight and composition

Body weight was measured 2×/week. Body composition including lean mass, fat mass, free water content, and total water content of non-anesthetized mice was measured weekly using EchoMRI-100 quantitative magnetic resonance whole body composition analyzer (Echo Medical Systems, Houston, TX).

## Vertical sleeve gastrectomy

To reduce bariatric surgery-associated weight loss, perioperative measures included providing liquid diet (Ensure Original Milk Chocolate Nutrition Shake, Abbott, Chicago, IL) and DietGel recovery (Clear H$_2$O, Portland, ME, ID# 72-06-5022) 1 day before surgery to all mice. At 4 hr before surgery, solid food was removed to reduce stomach contents. For 4 hr pre-surgery, mice were maintained half on half off a heat pad in clean new cages. Surgery was performed under isoflurane anesthesia. VSG was performed as previously described (*Yin et al., 2012*) with additional control dietary intervention for comparison of weight loss approaches. The stomach was clamped, and the lateral 80% of the stomach was removed with scissors. The remaining stomach was sutured with 8–0 to create a tubular gastric sleeve. All treatment groups not receiving VSG had a sham surgery performed. For sham, an abdominal laparotomy was performed with exteriorization of the stomach. Light pressure with forceps was applied to the exteriorized stomach. For both VSG and sham surgeries, the abdominal wall was closed with 6–0 sutures and skin closed with staples. Mice received carprofen (5 mg/kg, subcutaneous, once daily) as an analgesic immediately prior to and once daily for 3 days following surgery. Mice were given 1 mL saline at time of surgery. Perioperative procedures were performed in accordance with the literature (*Doerning et al., 2018*; *Stevenson et al., 2019*). For 12 hr post-surgery, mice were maintained half on half off a recovery heat pad. Mice were provided Ensure liquid diet (as above), DietGel recovery, and solid food pellets ad libitium for 48 hr post-surgery. HFD-fed DIO mice receiving VSG ('HFD-VSG') were maintained on the same HFD for 5 weeks following surgery until euthanasia at study endpoint (*Figure 1A*). Control groups that were lean ('LFD-Sham') or DIO ('HFD-Sham') were maintained on respective LFD or HFD diets following sham surgery. For dietary intervention weight loss, DIO mice received sham surgery and were subjected to weight loss intervention following sham surgery for 5 weeks until endpoint. 'Weight-matched' (WM) mice were controls to the HFD-VSG mice by weight matching through restricting intake of HFD (*Sipe et al., 2017*). On average, mice consumed 1.7 g (ranging from 1.0 to 2.5 g or 8.84 kcal [5.2–13.0 kcal]) per day of HFD. Mice were fed at the start of the dark cycle. 78.9% of VSG mice survived to endpoint (30/38).

## Tumor cell implantation

E0771 murine adenocarcinoma breast cancer cell line was originally isolated from a spontaneous tumor from C57BL/6 mouse. E0771 cells were purchased from ATCC (CRL-3461) and stable transfected to express luciferase (luc) (*Ouzounova et al., 2017*) by the Korkaya group at Augusta University (*Pingili*

*et al., 2021*; *Ouzounova et al., 2017*). Cells tested negative for mycoplasma (Lonza, Basel) and were cultured as described previously, cell identity verified by breast cancer subtype expression analysis (*Pingili et al., 2021*). Briefly, cells were cultured in RPMI containing 10% FBS, 100 U/mL of penicillin, and 100 µg/mL streptomycin in a humidified chamber at 37°C under 5% $CO_2$. E0771 cells were injected in the left fourth mammary fat pad of 22-week-old C57BL/6J females at 250,000 cells in 100 µL of 75% RPMI/25% Matrigel. When tumors became palpable (typically 1 week after implantation), tumor growth was monitored 2×/week by measuring the length and width of the tumor using digital calipers. Tumor volume was calculated using the following formula: volume = (width)$^2$ × (length)/2 (*Pingili et al., 2021*). No tumors failed to take, and tumor regression was not detected. At the endpoint on day 21 after tumor cell injection, excised tumor mass was determined.

## Immune checkpoint blockade

In a separate experimental cohort limited to HFD-VSG and controls including LFD-Sham, HFD-Sham, and WM-Sham, mice were subjected to the same dietary and surgical study design above (*Figure 1A*). After 20 weeks on LFD or HFD, 24-week-old mice received either a sham or VSG surgery. Two weeks following surgery, mice were injected with E0771-luc cells as above. ICB included anti-PD-L1 antibody (Clone 10 F.9G2, #BE0101) and IgG2b isotype control (Clone LTF-2, #BE0090), purchased from BioXcell (West Lebanon, NH). Antibody administration by intraperitoneal injection began 3 days after E0771 cell injection when tumors were palpable (width of >2.5 mm). Mice were injected every third day for 21 days until endpoint (8 mg/kg) (*Rigo et al., 2017*).

## Tissue and blood collection

Three weeks after tumor implantation (i.e. 5 weeks after surgery), mice were fasted for 4 hr and anesthetized. Blood was collected via cardiac puncture into EDTA-coated vials. Plasma was separated from other blood components by centrifugation at 1200×g for 45 min at 12°C. Mammary tumors, tumor adjacent mammary fat pad, unaffected inguinal mammary fat pad, and gonadal adipose were weighed and either flash frozen in liquid nitrogen, placed into a cassette and formalin-fixed, or digested into a single cell suspension for flow cytometry. All frozen samples were stored at −80°C until analyzed.

## Plasma adipokines and cytokines

Plasma collected at sacrifice was used for measuring leptin and IL-6 using the Milliplex MAP Mouse Metabolic Hormone Magnetic Bead Panel in the Luminex MAGPIX system (EMD Millipore, Billerica, MA).

## Flow cytometric analysis of tumors and adjacent mammary adipose tissue

Flow cytometry analysis was done as previously described (*Pingili et al., 2021*). In brief, excised tumors (200 mg) were dissociated in RPMI media containing enzyme cocktail mix from the mouse tumor dissociation kit (Miltenyi Biotec, Auburn, CA) and placed into gentleMACS dissociators per manufacturer's instructions. Spleen single cell suspensions were obtained by grinding spleens against 70 µm filter using a syringe plunger. Following red blood cell lysis (Millipore Sigma, St. Louis, MO), viability was determined by staining with Ghost dye (Tonbo Biosciences Inc) followed by FcR-blocking (Tonbo). Antibodies were titrated, and separation index was calculated using FlowJo v. 10 software. Cells were stained with fluorescently labeled antibodies and fixed in Perm/fix buffer (Tonbo). Stained cells were analyzed using Bio-Rad ZE5 flow cytometer. Fluorescence minus one (FMO) stained cells and single color Ultracomp Beads (Invitrogen, Carlsbad, CA) were used as negative and positive controls, respectively. Data was analyzed using FlowJo v. 10 software (Treestar, Woodburn, OR). Total immune cells from tumor and tumor adjacent mammary fat pad (including TdLN) were gated by plotting forward scatter area versus side scatter area, single cells by plotting side scatter height versus side scatter area, live cells by plotting side scatter area versus Ghost viability dye, and immune cells by plotting CD45 versus Ghost viability dye. T cells were gated as follows in tumor CD3+ T cells (CD3+) and CD8+ T cells (CD3+ and CD8+). Macrophages are gated as CD11b+ and F480+. M-MDSC are gated as CD11b+ Ly6C$^{high}$, Ly6G−. Non-immune cells were gated as CD45− and MFI for PD-L1. Gates were defined by FMO stained controls and verified by back-gating of cell populations. Gating schema is shown in *supplementary file 2*.

### Flow cytometric analysis of E0771 breast cancer cells

E0771-luc cells were treated with recombinant mouse IL-6 (200 pg/mL) for 4 hr. Representative biological replicate plotted with N=3 biological replicates with significance. Following trypsinization, cells were stained with Ghost dye (Tonbo Biosciences Inc) followed by FcR-blocking (Tonbo) and fluorescent PD-L1 antibody. Flow cytometry performed and analyzed as above for PD-L1 MFI.

### Tumor and mammary fat pad RNA-seq

mRNA was extracted from tumor tissue using RNeasy mini kit (QIAGEN, Germantown, MD) and mammary fat pad tissue using a kit specific for lipid rich tissue (Norgen Biotek, Ontario, Canada). The integrity of RNA was assessed using Agilent Bioanalyzer and samples with RIN >8.0 were used. Libraries were constructed using NEBNext Ultra RNA Library Prep Kits (non-directional) for Illumina, following manufacturer protocols. mRNA was enriched using oligo-dT beads. Libraries were sequenced on NovaSeq 6000 using paired-end 150 bp reads. There was no PhiX spike-in. Data was analyzed as described previously (*Pingili et al., 2021*; *Choi et al., 2021*). RNA-seq statistical differences between experimental groups were determined as described previously (*Pingili et al., 2021*). In brief, Benjamini-Hochberg procedure was used to control false discovery rate (FDR) for adjusted p value . RNA-seq data has been uploaded as GEO GSE174760, GSE174761, and GSE174762. Transcript-level abundance was imported into gene-level abundance with the R package tximport. Genes with low expression were identified and filtered out from further analysis using filterByExpr function of the edgeR package in R software. Voom transformation function was applied to normalize log2-cpm values using mean-variance trend in the limma software package. ClaNC was used to create classifier genes that characterize the groups of interest for semi-supervised heatmaps. DAVID v6.8 was used for pathway analysis and enriched pathways defined as an FDR less than 10 percent (*Huang et al., 2009*). Immune infiltration estimations based on bulk gene expression data from RNA-seq were plotted using TIMER2.0 (*Li et al., 2020a*) and CIBERSORT (*Newman et al., 2015*).

### Bariatric surgery patient RNA-seq

Patient gene expression from subcutaneous adipose tissue pre- and post-bariatric surgery was downloaded from GSE65540 (*Poitou et al., 2015*), and counts were normalized using counts per million. EdgeR was used for differential expression analysis, and significance was defined as adjusted p value of <0.1. Benjamini-Hochberg was used to calculate the FDR. Mouse and human Venn diagram was created using the interactive Venn website.

### Gene expression

Total RNA was isolated from tumors and reversed transcribed to cDNA using High-Capacity cDNA Reverse Transcription Kit (Applied Biosystems). Quantitative RT-PCR was performed with iTaq Universal SYBR Green Supermix (Bio-Rad). Primers span an exon-exon junction and were designed with Primer-BLAST (NCBI). Relative gene expression was calculated normalized to 18S transcript with $2^{-\Delta\Delta Ct}$. Primer sequences are:

> *Ifng* F:GGATGCATTCATGAGTATTGC, *Ifng* R:GTGGACCACTCGGATGAG,
> *Prf1* F:GAGAAGACCTATCAGGACCA, *Prf1* R:AGCCTGTGGTAAGCATG,
> *Gzmb* F:CCTCCTGCTACTGCTGAC, *Gzmb* R:GTCAGCACAAAGTCCTCTC,
> *18S* F:TTCGGAACTGAGGCCATGATT, and *18S* R:TTTCGCTCTGGTCCGTCTTG

### Histology and quantification

Tumors and normal fourth mammary fat pads (contralateral to the injected tumor bearing mammary fat pad) were isolated at the time of sacrifice and fixed in 10% formalin. Formalin fixed paraffin embedded (FFPE) sections from tumors and adipose were cut at 5 µm thickness. FFPE sections were stained with hematoxylin and eosin and scanned by Thermo Fisher (Panoramic 250 Flash III, Thermo Fisher, Tewksbury, MA) scanner, and adipocyte area of N=50 adipocytes were quantified using software (Case Viewer) along the longest diameter per adipocyte.

## Statistics

Statistical differences between experimental groups were determined using one-way or two-way ANOVA (as noted in figure legends) with Fisher's LSD test for individual comparisons. Outliers were identified and excluded based on the ROUT method with Q=1%. For body weight, body composition, and tumor volume over time within animals, data was treated as repeated measures. All statistics were performed using statistical software within Graphpad Prism (Graphpad Software, Inc, La Jolla CA). All data are shown as mean ± SEM. p Values less than 0.05 were considered statistically significant. Sample size was determined by power analysis calculations and pilot experiments. Group allocation was done to ensure equal distribution of starting body weight between groups.

## Study approval

Animal studies were performed with approval and in accordance with the guidelines of the IACUC at the University of Tennessee Health Science Center and in accordance with the National Institutes of Health Guide for the Care and Use of Laboratory Animals.

## Acknowledgements

We acknowledge support from the following funding sources: National Institutes of Health grant NCI R01CA253329 (LM, JFP, MJD); National Institutes of Health grant NCI R37CA226969 (LM, DNH); The Mary Kay Foundation (LM); V Foundation (LM, DNH); National Institutes of Health grant NIDDK R01DK127209 (JFP); Tennessee Governor Pediatric Recruitment Grant (JFP); Tennessee Clinical and Translational Science Institute (JFP); Transdisciplinary Research on Energetics and Cancer R25CA203650 (LMS); American Association for Cancer Research Triple Negative Breast Cancer Foundation Research Fellowship (LMS); National Institutes of Health grant NCI F32 CA250192 (LMS); The Obesity Society/Susan G Komen Cancer Challenge award 2018 (LMS); National Institute of Health grant NCI F30CA265224 (JRH); National Institute of Health grant NCI U24CA210988 (DNH); National Institute of Health grant NCI UG1CA233333 (DNH); National Institute of Health grant NCI R01CA121249 (JAC); UT/West Cancer Research Institute Fellowship 2019 (JCC); NIH Medical Student Research Fellowship (MSRF) Program 2019 (NAJ); We thank Daniel Johnson from UTHSC Molecular Resource Center.

## Additional information

### Funding

| Funder | Grant reference number | Author |
| --- | --- | --- |
| National Cancer Institute | R01CA253329 | Matthew J Davis<br>Joseph F Pierre<br>Liza Makowski |
| National Cancer Institute | R37CA226969 | D Neil Hayes<br>Liza Makowski |
| National Cancer Institute | F32 CA250192 | Laura M Sipe |
| National Cancer Institute | R25CA203650 | Laura M Sipe |
| Mary Kay Foundation | 05-20 | Liza Makowski |
| V Foundation for Cancer Research | | D Neil Hayes<br>Liza Makowski |
| National Institute of Diabetes and Digestive and Kidney Diseases | R01DK127209 | Joseph F Pierre |
| American Association for Cancer Research | Triple Negative Breast Cancer Foundation Research Fellowship | Laura M Sipe |
| National Cancer Institute | F30CA265224 | Jeremiah R Holt |

| Funder | Grant reference number | Author |
| --- | --- | --- |
| National Cancer Institute | CA262112 | Liza Makowski<br>D Neil Hayes |

The funders had no role in study design, data collection and interpretation, or the decision to submit the work for publication.

## Author contributions

Laura M Sipe, Conceptualization, Formal analysis, Funding acquisition, Investigation, Methodology, Project administration, Supervision, Writing - original draft, Writing – review and editing; Mehdi Chaib, Formal analysis, Investigation, Methodology, Writing – review and editing; Emily B Korba, Mary Camille Lovely, Investigation, Writing – review and editing; Heejoon Jo, Ubaid Tanveer, Jeremiah R Holt, Radhika Sekhri, Formal analysis; Brittany R Counts, Margaret S Bohm, Investigation; Jared C Clements, Neena A John, Deidre Daria, Tony N Marion, Ajeeth K Pingili, Bin Teng, Methodology; James A Carson, Resources, Supervision; D Neil Hayes, Funding acquisition, Resources, Supervision, Writing – review and editing; Matthew J Davis, Supervision, Writing – review and editing; Katherine L Cook, Conceptualization, Funding acquisition, Writing – review and editing; Joseph F Pierre, Conceptualization, Funding acquisition, Methodology, Resources, Supervision, Writing – review and editing; Liza Makowski, Conceptualization, Funding acquisition, Methodology, Project administration, Resources, Supervision, Writing - original draft, Writing – review and editing

## Author ORCIDs

Laura M Sipe  http://orcid.org/0000-0001-7848-3848
Emily B Korba  http://orcid.org/0000-0001-7422-9084
Joseph F Pierre  http://orcid.org/0000-0002-4248-1290
Liza Makowski  http://orcid.org/0000-0002-5337-8037

## Ethics

Animal studies were performed with approval and in accordance with the guidelines of the Institutional Animal Care and Use Committee (IACUC) at the University of Tennessee Health Science Center (Animal Welfare Assurance Number A3325-01) and in accordance with the National Institutes of Health Guide for the Care and Use of Laboratory Animals . The protocol was approved under the protocol identifier 21.0224.

## Decision letter and Author response

Decision letter https://doi.org/10.7554/eLife.79143.sa1
Author response https://doi.org/10.7554/eLife.79143.sa2

# Additional files

## Supplementary files

• Supplementary file 1. Detailed multiple comparisions for body weight. (a) Multiple comparisons of body weight after surgery over time. *p<0.05, **p<0.01, ***p<0.001, and ****p<0.0001. Two-way ANOVA with Fisher's LSD test. Low fat diet (LFD), high fat diet (HFD), vertical sleeve gastrectomy (VSG), and weight-matched (WM). (b) Multiple comparisons of tumor volume over time. *$P$<0.05, **$P$<0.01, ***$P$<0.001, ****$P$<0.0001. Two-Way ANOVA with Fisher's LSD test. LFD, HFD, VSG, and WM. (c) Conserved differentially expressed genes in subcutaneous adipose/mammary fat pad in obese and bariatric surgery patients and mice.

• Supplementary file 2. Gating schema for flow cytometric analysis of immune cells in tumor single cell suspensions. Total cells from tumor or tumor adjacent mammary fat pad (including tumor draining lymph node, TdLN) were gated by plotting forward scatter area versus side scatter area, single cells by plotting side scatter height versus side scatter area, live cells by plotting side scatter area versus Ghost viability dye, and immune cells by plotting CD45 versus Ghost viability dye. T cells were gated as follows: CD3+ T cells (CD3+) and CD8 + T cells (CD3+ and CD8+). Mean fluorescent intensity (MFI) of PD-1 was measured in CD3+ PD-1+ cells. Monocytic myeloid derived suppressor cells (M-MDSC) are gated as CD11b+, Ly6C$^{high}$, and Ly6G−. Macrophages are gated as CD11b+ and F480+. Non-immune cells were gated as CD45−, PD-L1+, and MFI for PD-L1.

• MDAR checklist

## Data availability

The data generated in this study are available within the source data file stored in Dryad Digital Repository, doi:10.5061/dryad.w0vt4b8tq. The RNA-seq data generated in this study are publicly available in NCBI GEO GSE174760 of tumor RNA-seq and NCBI GEO GSE174761 of mammary fat pad RNA-seq.

The following datasets were generated:

| Author(s) | Year | Dataset title | Dataset URL | Database and Identifier |
|---|---|---|---|---|
| Makowski L, Pierre JF, Hayes DN, Sipe LM | 2021 | A5738 | https://www.ncbi.nlm.nih.gov/geo/query/acc.cgi?acc=GSM5326659 | NCBI Gene Expression Omnibus, GSM5326659 |
| Makowski L, Pierre JF, Hayes DN, Sipe LM | 2021 | Bariatric surgery reduces obesity associated breast cancer and enhances response to immunotherapy [Tumor - breast cancer RNA-seq] | https://www.ncbi.nlm.nih.gov/geo/query/acc.cgi?acc=GSE174761 | NCBI Gene Expression Omnibus, GSE174761 |
| Sipe L, Chaib M, Korba E, Lovely M, Jo H, Counts B, Tanveer U, Clements J, John N, Daria D, Marion T, Sekhri R, Pingili A, Teng B, Carson J, Hayes D, Davis M, Pierre J, Makowski L | 2022 | Response to immune checkpoint blockade improved in pre-clinical model of breast cancer after bariatric surgery | https://doi.org/10.5061/dryad.w0vt4b8tq | Dryad Digital Repository, 10.5061/dryad.w0vt4b8tq |
| Makoski L, Pierre JF, Hayes DN, Sipe LM | 2021 | Bariatric surgery reduces obesity associated breast cancer and enhances response to immunotherapy [Mammary Fat Pad RNA-seq] | https://www.ncbi.nlm.nih.gov/geo/query/acc.cgi?acc=GSE174760 | NCBI Gene Expression Omnibus, GSE174760 |

The following previously published dataset was used:

| Author(s) | Year | Dataset title | Dataset URL | Database and Identifier |
|---|---|---|---|---|
| Tiret L, Perret C, Mathieu F, Truong V, Durand H, Ninio E, Poitou C, Clément K, Alili R, Chelghoum N, Torcivia A | 2015 | Impact of bariatric surgery on RNA-seq gene expression profiles of adipose tissue in humans | https://www.ncbi.nlm.nih.gov/geo/query/acc.cgi?acc=GSE65540 | NCBI Gene Expression Omnibus, GSE65540 |

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
