## [Editor Report]

This study investigates how weight loss by bariatric surgery or weight-matched dietary intervention impairs breast cancer growth as well as immunotherapy. This study can potentially provide some therapeutic intervention strategies on combining vertical sleeve gastrectomy and immunotherapy in treating breast cancer.

---

## [Decision Letter]

**Decision letter after peer review:**

Thank you for submitting your article "Response to immune checkpoint blockade improved in pre-clinical model of breast cancer after bariatric surgery" for consideration by *eLife*. Your article has been reviewed by 3 peer reviewers, including Qing Zhang as Reviewing Editor and Reviewer #1 and the evaluation has been overseen by W Kimryn Rathmell as the Senior Editor.

Essential revisions:

There are a few essential experiments that need to be performed as described below.

1) This study needs to use more than one syngeneic model to solidify the conclusion.

2) The detailed molecular mechanism of T cell signaling or Cytolysis in HFS VSG mice vs WM sham mice needs to be examined.

3) This study needs detailed characterization of which inflammatory response factors may mediate the phenotype of HFS VSG mice when compared to WM Sham mice. The data presented is currently mainly limited to RNA-Seq data, which lacks detailed characterization.

*Reviewer #2 (Recommendations for the authors):*

Introduction lacks a premise. The knowledge of the effects of bariatric surgery on cancer risk and progression needs to be introduced and cited. Discussion also needs to dissect the implications of the findings for cancer prevention and intervention.

The discovered BSAS gene expression signature and clinical findings should be covered in the Abstract.

Non-immune cells need to be defined better: malignant and/or vascular/stromal?

The authors attempted to investigate if it is the diet or the amount of adipose tissue that matters by comparing LFD-sham and WM-sham groups. The conclusion appears to be that it is adiposity rather than the diet that matters for tumor growth and ICB response. This needs to be emphasized in Discussion and Abstract.

Examining the potentiation of ICB by weight loss through diet caloric restriction (WM-Sham group) is an important final piece of data making the story complete. Could it be speculated why HFD-VSG tumors upon ICB still grew as large as in untreated LFD-sham and WM-sham groups in Fig 5?

*Reviewer #3 (Recommendations for the authors):*

1. The lack of use of a spontaneous model of breast cancer for these studies, such as the widely used MMTV-PyMT (mouse mammary tumor virus-polyoma middle tumor-antigen) is a major concern. These mice develop spontaneous mammary tumors that closely resemble the different stages of tumor progression of human breast cancers.

2. The authors should investigate and discuss the potential mechanisms underlying the increased expression of PD-L1 on tumor cells post-VSG.

3. For the proof of concept, the authors should use a PD-L1 knockout E0771 cell line for these studies.

4. To specifically distinguish the source of PD-L1 in the tumor microenvironment, the authors should have used the epithelial cell adhesion molecule (EpCAM, CD326), which is expressed by the cancer cells.

5. Did the authors analyze the expression of PD-L1 on other immune cells especially macrophages as increased PD-L1 in macrophages has been shown to correlate with high levels of PD-L1 in tumors (PMID: 31615933).

6. Did the authors analyze the influence of bariatric surgery on distant metastasis? Why T cell infiltration is low in tumors post-VSG.

7. The experiments are very briefly described in the Results section.

---

## [Author Response]

Essential revisions:There are a few essential experiments that need to be performed as described below.1) This study needs to use more than one syngeneic model to solidify the conclusion.

We have addressed this below in response to reviewer 3 question 1.

2) The detailed molecular mechanism of T cell signaling or Cytolysis in HFS VSG mice vs WM sham mice needs to be examined.

We have addressed this below in response to reviewer 1 question 2 and reviewer 3 question 5.

3) This study needs detailed characterization of which inflammatory response factors may mediate the phenotype of HFS VSG mice when compared to WM Sham mice. The data presented is currently mainly limited to RNA-Seq data, which lacks detailed characterization.

We have addressed this below in response to reviewer 1 question 1.

Reviewer #2 (Recommendations for the authors):Introduction lacks a premise. The knowledge of the effects of bariatric surgery on cancer risk and progression needs to be introduced and cited. Discussion also needs to dissect the implications of the findings for cancer prevention and intervention.

Thank you for your review. We have added a premise to the introduction. We also agree about the importance of bariatric surgery on breast cancer risk and progression, therefore additional text has been added to the introduction and discussion. We have also included the new JAMA study published June 3, 2022, entitled “June 3, 2022 Association of Bariatric Surgery With Cancer Risk and Mortality in Adults With Obesity” which further supports interesting protective effects post bariatric surgery.

The discovered BSAS gene expression signature and clinical findings should be covered in the Abstract.

We have added this the Abstract.

Non-immune cells need to be defined better: malignant and/or vascular/stromal?

This is an important question that we cannot answer for this manuscript. We did not examine the CD45- fraction in further detail but intend to do so in the future, which is beyond the scope of this study. However, we have added additional supportive flow cytometric analysis in response to reviewer 1 demonstrating that in addition to the CD45- fraction demonstrating elevated PD-L1 in the HFD-VSG tumors, PD-L1 was also elevated on tumor associated macrophages as well as PD-L1 M-MDSCs in tumors which are now included as a new figures 4G-H. We have included details in the methods, results, and added citations and discussion about PD-L1 positivity on M-MDSCs and macrophages in the discussion which we believe strengthens the manuscript.

The authors attempted to investigate if it is the diet or the amount of adipose tissue that matters by comparing LFD-sham and WM-sham groups. The conclusion appears to be that it is adiposity rather than the diet that matters for tumor growth and ICB response. This needs to be emphasized in Discussion and Abstract.

Thank you for your comments. We have addressed this in our edits by adding to the discussion that “Thus, in mice from the VSG group and WM control groups, results suggest that tumor responses aligned with adiposity not diet exposure. Both groups were fed the same high fat diet as obese mice which presented with the greatest adiposity and largest tumors suggesting that diet per se is not as important as adiposity.”

However, the answer to this question is not so clear cut for the response between the VSG and the weight matched sham control. The gonadal and tumor adjacent adipose fat pad weights were the same in both the primary and the ICB study. Adiposity typically associates with adipose size and circulating leptin, but we reported in Fig 2D-F that these factors were different between HFD-VSG and WM-sham.

Furthermore, the ICB study shows greater efficacy after VSG while the mice in the WM Sham group showed moderate and not significantly improved efficacy (perhaps because as noted tumors were already small). Both groups had similar adiposity, and both were on high fat diet. Thus, we believe that underlying immune related responses such as those presented herein (new findings on IL-6 and PD-L1 after VSG) or those to be discovered contribute to response to ICB.

Examining the potentiation of ICB by weight loss through diet caloric restriction (WM-Sham group) is an important final piece of data making the story complete. Could it be speculated why HFD-VSG tumors upon ICB still grew as large as in untreated LFD-sham and WM-sham groups in Fig 5?

Thank you for your comment. Tumors in the VSG- treated group developed tumors that matched lean IgG-treated controls as reviewer 2 points out. The failure of anti-PD-L1 ICB after HFD-VSG to reduce progression even further compared to lean controls might be due to perhaps dosing or timing of experimental intervention. In addition, it is possible that a combination of a chemotherapy + ICB could entirely blunt tumor progression. We have added these comments to our discussion.

Reviewer #3 (Recommendations for the authors):1. The lack of use of a spontaneous model of breast cancer for these studies, such as the widely used MMTV-PyMT (mouse mammary tumor virus-polyoma middle tumor-antigen) is a major concern. These mice develop spontaneous mammary tumors that closely resemble the different stages of tumor progression of human breast cancers.

Thank you for your questions. Your question is important and throughout our studies, we have thought about how to address this concern using a complementary female model for breast cancer. There are many limitations and hurdles to examine obesity, breast cancer, and weight loss in female mice which makes the model and our results presented herein important.

First, the cancer model MMTV-PyMT suggested by reviewer 3 is unfortunately not likely to yield interpretable results. In the C57Bl/6J E0771 model we used, female mice were on a high fat diet for 16 weeks in order to generate diet induced obesity. In the BL6 background, the MMTV-PyMT latency is 12 weeks of age. This time course is too short to have HFD-induced obesity and subsequent surgically-induced weight loss before tumor growth.

Second, commonly used models for breast cancer such as 4T1 or EMT6 are derived from BALB/c mice. BALB/c mice are resistant to diet-induced obesity, hence cannot be used for studies on obesity and weight loss.

Third, is a consideration on the extensive burden and cost to repeat such a study. We chose the C57Bl/6J model to study obesity and bariatric surgery in breast cancer because this strain is the most obesogenic and is widely accepted in the obesity field. To study obesity in females, it takes 4 months of high fat diet exposure to generate mice considered obese. An approach to accelerate obesity in females is to conduct an ovariectomy, subjecting the mice to two separate surgeries (first ovariectomy then VSG), which was not feasible. Therefore, each study takes many months for female mice to develop diet induced obesity and then recover from bariatric surgery or sham surgical controls well before the tumor studies begin.

Last, these experiments take specialized training. Since our goal was to examine obesity, mice could not lose weight from the stress of surgery. Hence great care of each mouse must be maintained and was extensively optimized to have a highly successful surgery and survival. Because pre- and post-care as well as the surgeries are very labor intensive, cohorts are randomized and staggered, making these highly coordinated and laborious experiments for staff (including vets) and postdoctoral and student trainees.

In sum, to date, there are no publications on bariatric surgery and breast cancer nor on bariatric surgery and breast cancer with immunotherapy using pre-clinical models. To address your valid concern, we have noted in the discussion that few models exist to study both highly obesogenic strains and genetic or syngeneic breast cancer. Overall, we hope the reviewer will agree with the novelty of the observations and relevance to the field of breast cancer immunotherapy of this large effort.

2. The authors should investigate and discuss the potential mechanisms underlying the increased expression of PD-L1 on tumor cells post-VSG.

We have now included a new figure demonstrating elevated plasma IL-6 as a potential mechanism to impact PD-L1. IL-6 is established to increase PD-L1 stability and further show IL-6 increases extracellular PD-L1 levels new figures 3 E and F.

3. For the proof of concept, the authors should use a PD-L1 knockout E0771 cell line for these studies.

This is an important question because PD-L1 is expressed on adipocytes, myeloid cells, and cancer cells, etc. We do not know if the elevated PD-L1 is on the cancer cells per se or other cells of the CD45- fraction. We have now clarified in our discussion that we do not posit that the increase in PD-L1 after VSG is specific to cancer cells, but instead on many cell types throughout the tumor microenvironment. In this manuscript, we opted to use anti-PD-L1 immunotherapy as a loss of function approach to block PD-L1 on as many cells as possible to best mimic clinical approaches. We have also provided new data in response to reviewer 1 question 2 and your question #5 below demonstrating elevated PD-L1 on M-MDSCs and macrophages in tumors from the HFD-VSG group as well.

4. To specifically distinguish the source of PD-L1 in the tumor microenvironment, the authors should have used the epithelial cell adhesion molecule (EpCAM, CD326), which is expressed by the cancer cells.

This is a great idea and exploration of the cancer cells via EpCAM/CD236 was a missed opportunity in our flow panel. PD-L1 is expressed on many cells throughout the tumor environment including adipocytes, myeloid cells (see response to your question 5), and cancer cells, etc. We surmise that the increase in PD-L1 after VSG is detected on many cell types throughout the tumor microenvironment. We addressed your concern below by adding additional information after examining macrophages as noted in reviewer 1 question 1 response.

5. Did the authors analyze the expression of PD-L1 on other immune cells especially macrophages as increased PD-L1 in macrophages has been shown to correlate with high levels of PD-L1 in tumors (PMID: 31615933).

This is an important point that we have now included as new figure in the manuscript (Figure 4G-H – see figures in response to reviewer 1 question 2 as well). Indeed, macrophages in tumors after VSG display a significantly greater frequency of PD-L1+ Macrophages in the HFD-VSG group. We also examined monocytic-MDSC (M-MDSC) PD-L1+ frequency and demonstrated significantly elevated PD-L1+ M-MDSCs in the HFD-VSG group compared to all other groups. These novel findings in addition to the CD45- PD-L1 positivity after VSG could account for greater efficacy with anti-PDL1 ICB. We have now included these new details in methods, results, and added citations and discussion about PD-L1 positivity on M-MDSCs and macrophages in the discussion. We also discussed and cited the manuscript that you mentioned [PMID: 31615933 https://pubmed.ncbi.nlm.nih.gov/31615933/] which demonstrated that tumors with high PD-L1 is associated with longer overall survival in patients treated with ICB. Thank you for your suggestions.

6. Did the authors analyze the influence of bariatric surgery on distant metastasis? Why T cell infiltration is low in tumors post-VSG.

We did not measure metastasis in this model. To address your question about why T cell infiltration is low in tumors post-VSG, we considered published work that we had cited showing that obesity decreased CD8^+^ cytotoxic tumor infiltrating T cells (Wang, Aguilar et al., 2019, Pingili, Chaib et al., 2021) and wondered how bariatric surgery-induced weight loss would impact cytotoxic T cells. We regret that we were not clearer in our manuscript. In Figure 4 A-C we reported that T cell frequency in tumors was lowest after VSG compared to all other diet and surgery groups. In contrast, T cell content in tumor draining lymph node and tumor adjacent mammary fat pad CD3+ and CD8^+^ T cell content was not changed (supplemental figure 1 A-B). We showed that T cell signaling and activation was lower post-VSG compared to W-Sham (Figure 4 E-F). Therefore, to address your concern, we have added additional discussion stating that it is likely that the elevated PD-L1 after VSG (in the CD45- fraction, and newly reported findings in macrophages and M-MDSCs) led to reduced T cell signaling, activation, which would lead to reduced CD3+ and CD8^+^ T cell content.

7. The experiments are very briefly described in the Results section.

Excuse our brevity. We have expanded our discussion of experimental approach in the Results section for more clarity.

Chan, L.-C., C.-W. Li, W. Xia, J.-M. Hsu, H.-H. Lee, J.-H. Cha, H.-L. Wang, W.-H. Yang, E.-Y. Yen, W.-C. Chang, Z. Zha, S.-O. Lim, Y.-J. Lai, C. Liu, J. Liu, Q. Dong, Y. Yang, L. Sun, Y. Wei, L. Nie, J. L. Hsu, H. Li, Q. Ye, M. M. Hassan, H. M. Amin, A. O. Kaseb, X. Lin, S.-C. Wang and M.-C. Hung (2019). "IL-6/JAK1 pathway drives PD-L1 Y112 phosphorylation to promote cancer immune evasion." Journal of Clinical Investigation 129(8): 3324-3338.

Li, Z., C. Zhang, J.-X. Du, J. Zhao, M.-T. Shi, M.-W. Jin and H. Liu (2020). "Adipocytes promote tumor progression and induce PD-L1 expression via TNF-α/IL-6 signaling." Cancer Cell International 20(1).

Pingili, A. K., M. Chaib, L. M. Sipe, E. J. Miller, B. Teng, R. Sharma, J. R. Yarbro, S. Asemota, Q. Al Abdallah, T. S. Mims, T. N. Marion, D. Daria, R. Sekhri, A. M. Hamilton, M. A. Troester, H. Jo, H. Y. Choi, D. N. Hayes, K. L. Cook, R. Narayanan, J. F. Pierre and L. Makowski (2021). "Immune checkpoint blockade reprograms systemic immune landscape and tumor microenvironment in obesity-associated breast cancer." Cell Reports 35(12): 109285.

Wang, Z., E. G. Aguilar, J. I. Luna, C. Dunai, L. T. Khuat, C. T. Le, A. Mirsoian, C. M. Minnar, K. M. Stoffel, I. R. Sturgill, S. K. Grossenbacher, S. S. Withers, R. B. Rebhun, D. J. Hartigan-O’Connor, G. Méndez-Lagares, A. F. Tarantal, R. R. Isseroff, T. S. Griffith, K. A. Schalper, A. Merleev, A. Saha, E. Maverakis, K. Kelly, R. Aljumaily, S. Ibrahimi, S. Mukherjee, M. Machiorlatti, S. K. Vesely, D. L. Longo, B. R. Blazar, R. J. Canter, W. J. Murphy and A. M. Monjazeb (2019). "Paradoxical effects of obesity on T cell function during tumor progression and PD-1 checkpoint blockade." Nature Medicine 25(1): 141-151.